# FairPATE: Exposing the Pareto Frontier of Fairness, Privacy, Accuracy, and Coverage

## Abstract

Deploying machine learning (ML) models often requires both fairness and privacy guarantees. In this work, we study the challenges of integrating group fairness interventions into the Private Aggregation of Teacher Ensemble (PATE) framework. We show that in the joint fairness-privacy setting, the placement of the fairness intervention before, or after PATE's noisy aggregation mechanism (which ensures its differential privacy guarantees) leads to excessive fairness violations, or inefficient privacy budgeting, respectively. With this in mind, we present FairPATE which adds a rejection mechanism due to fairness violations. Through careful adjustment of PATE's privacy accounting, we match the DP-SGD-based state-of-the-art privacy-fairness-accuracy trade-offs (Lowy et al., 2023) in demographic parity, and improve on them for equality of odds with 2% lower disparity at similar accuracy levels and privacy budgets. We also evaluate FairPATE in the setting where exact fairness guarantees need to be enforced by refusing to provide algorithmic decisions at inference-time (for instance, in a human-in-the-loop setting) thus trading off fairness with coverage. Based on our FairPATE, we provide, for the first time, empirical Pareto frontiers for fairness, privacy, accuracy and coverage on a range of privacy and fairness benchmark datasets.

## 1 Introduction

From medical applications (Irvin et al., 2019) to infrastructure planning with census data (US Census Bureau, 2020), deploying machine learning (ML) models in critical contexts often requires not only utility (accuracy) guarantees, but also fairness and privacy assurances. Differential Privacy (DP) is the *de facto* standard for privacy-preserving ML. In this work, we highlight three main challenges of integrating group fairness interventions in Private Aggregation of Teacher Ensemble (PATE)—one of the two canonical privacy-preserving ML frameworks. PATE takes advantage of privacy-preserving transfer learning between teacher and student models; and as a result, provides multiple possible interventions points to preserve fairness. However, not all such interventions are equally effective.

We find that when optimizing for both privacy and fairness, instead of locating the fairness intervention with respect to the training *process*—as is commonly done in the algorithmic fairness literature—it is more beneficial to locate it with respect to the *differential privacy (DP) intervention*. In PATE, this DP intervention takes the form of a private labeling process with a query rejection mechanism. Next, we highlight three challenges for placing a fairness intervention in PATE. First, we argue that pre-DP interventions where a fairness constraint is placed before DP noising can be ineffective in satisfying the constraint. Second, we theoretically show a particular instance of such interventions—one that balances class labels for different subpolulations—leads to degraded privacy guarantees. Third, we argue that post-DP interventions in PATE, suffer from inaccurate privacy budgeting.

To address these challenges, we present FairPATE which augments PATE's query rejection mechanism with a fairness constraint. To make DP accounting accurate, we adjust PATE privacy accounting to consider the possibility of rejecting queries due to fairness. We achieve state-of-the-art (SOTA) privacy-fairness-accuracy trade-offs, improving on the previous SOTA (Lowy et al., 2023) under equality of odds fairness notion by up to 2% improvement in average disparity at similar accuracy levels over a range of privacy budgets.

We initially present FairPATE assuming a demographic parity notion of fairness since PATE relies on unlabeled public data, which lacks the ground truth labels $Y$ necessary to estimate other group fairness metrics (such as *equality of odds*, or *loss disparity*). To enable the use of metrics that rely on labels, we present a novel fairness metric estimation where we decompose each metric into components which include $Y$, and those that do not. Using a small dataset with ground truth labels we calculate the former in an offline fashion and once for every dataset. This allows us to estimate standard group fairness (demographic parity, equality of odds, error parity, etc.) metrics at inference time for the unlabeled dataset.

Finally, we consider FairPATE within a larger decision making context where producing unfair decisions is much more costly than refusing to answer. For instance, consider a human-in-the-loop system used for bail decisions . If at inference-time a decision cannot be made without violating a pre-specified fairness metric, then the model can refuse to answer, at which point that decision could be relegated to a human judge. From a technical point of view, this relaxes the classification setting to that of *classification with a reject option* with the difference being coverage is trading off with fairness instead of accuracy. We implement this setting using a inference-time post-processor that mirrors (but is independent of) our new aggregation algorithm for FairPATE. Empirically, we find that once the reject option is activated (*i.e.*, when a tighter fairness gurantee is desired than is achievable via fairness intervention) privacy, fairness and accuracy are no longer in a trade-off with each other.

We run an extensive empirical evaluation in several domains over multiple tabular (Adult, Retired-Adult, Parkinson's, Credit-Card) and vision (UTKFace, FairFace, CelebA, CheXpert) datasets, and provide, for the first time to the best of our knowledge, empirical Pareto frontiers for fairness, privacy, accuracy and coverage.

## 2 BACKGROUND

We denote the ML model for classification by $\theta$, the features as $(\mathbf{x}, z) \in \mathcal{X} \times \mathcal{Z}$ where $\mathcal{X}$ is the domain of non-sensitive attributes, $\mathcal{Z}$ is the domain of the sensitive attribute (categorical variable). The categorical class-label is denoted by $y \in [1, \ldots, K]$. We refer the interested reader to Appendix G for a more thorough overview.

**Group Fairness.** We base our work on the fairness metric of *multi-class demographic parity* which requires that ML models produce similar success rates (*i.e.*, rate of predicting a desirable outcome, such as getting a loan) for all sub-populations (Calders & Verwer, 2010). In practice, we estimate multi-class demographic disparity for class $k$ and subgroup $z$ with: $\widehat{\Gamma}(z, k) := \frac{|\{\hat{Y}=k, Z=z\}|}{|\{Z=z\}|} - \frac{|\{\hat{Y}=k, Z\neq z\}|}{|\{Z\neq z\}|}$, where $\hat{Y} = \theta(\mathbf{x}, z)$. We define demographic *parity* when the worst-case demographic disparity between members and non-members for any subgroup, and for any class is bounded by $\gamma$:

**Definition 1** ($\gamma$-DemParity). *For predictions $Y$ with corresponding sensitive attributes $Z$ to satisfy $\gamma$-bounded demographic parity ($\gamma$-DemParity), it must be that for all $z$ in $\mathcal{Z}$ and for all $k$ in $\mathcal{K}$, the demographic disparity is at most $\gamma$: $\Gamma(z, k) \leq \gamma$.*

**Differential Privacy.** Differential privacy Dwork & Roth (2013) is a framework to protect privacy of individuals when analyzing their data. It achieves this by *adding controlled noise* to the algorithm used for analysis, making it difficult to identify individual contributions while still providing useful statistical results. More formally, $(\varepsilon, \delta)$-differential privacy can be expressed as follows:

**Definition 2** ($(\varepsilon, \delta)$-Differential Privacy). *Let $\mathcal{M} \colon \mathcal{D}^* \to \mathcal{R}$ be a randomized algorithm that satisfies $(\varepsilon, \delta)$-DP with $\varepsilon \in \mathbb{R}_+$ and $\delta \in [0, 1]$ if for all neighboring datasets $D \sim D'$, i.e., datasets that differ in only one data point, and for all possible subsets $R \subseteq \mathcal{R}$ of the result space it must hold that $\mathbb{P}\left[\mathcal{M}(D) \in R\right] \leq e^\varepsilon \cdot \mathbb{P}\left[\mathcal{M}(D') \in R\right] + \delta$.*

**Private Aggregation of Teacher Ensemble (PATE).** As illustrated in Figure 1a, PATE (Papernot et al., 2016) trains an ensemble of so-called *teacher models* on disjoint subsets of the private data without any privacy protection. As a result, these teachers cannot be publicly exposed because they would leak information about the private data. Instead, PATE utilizes them to label a public dataset within an appropriately noised knowledge transfer process. Therefore, the teachers in the ensemble each vote for a label for each public data point, and the final label is determined as a noisy

$\arg\max$ over the teachers' vote. PATE estimates the privacy cost of answering queries (*i.e.* labeling data) through *teachers consensus* with higher consensus revealing less information about individual teachers, and, thereby, consuming less privacy costs. To take advantage of the fact that estimating consensus is less privacy-costly than answering queries, PATE rejects high-cost queries to save on the privacy budget (see Algorithm 4 in Appendix G.2). Both consensus estimation and vote aggregation (answering the query) are noised with $\mathcal{N}(0, \sigma_1^2)$ and $\mathcal{N}(0, \sigma_2^2)$, respectively; where $\sigma_1, \sigma_2$ are tuned for better final utility of the separate *student model* that is then trained on the labeled public dataset. The student can eventually be deployed publicly while the teachers are never externally exposed.

## 3 CHALLENGES OF UNFAIRNESS MITIGATION IN PATE

Group fairness interventions (including demographic parity) in ML are often categorized with respect to the training *process* as either pre-processing, in-processing, or post-processing. PATE's semi-supervised design, with both teacher and student models, then allows for multiple intervention points to implement unfairness mitigation (see Figure 1a). Designs with Teacher-level interventions are ❶ PATE-$T_{pre}$, ❷ -$T_{in}$, and ❸ -$T_{post}$. Corresponding Student-level interventions are ❺ PATE-$S_{pre}$, ❻ -$S_{in}$, and ❼ -$S_{post}$. To motivate the choice of intervention point for our FairPATE in Section 4, we analyze the other intervention points and highlight three main challenges that they suffer from.

**(C1) DP-Induced Fairness Violations.** Assume a non-private model $\theta : \mathcal{X} \mapsto \mathcal{Y}$ that satisfies a fairness constraint $C$. If we privatized $\theta$ by adding sensitivity-calibrated noise to its predictions, and thus create model $\theta'$; $\theta'$ is not guaranteed to satisfy $C$ anymore. Prior work has also observed this phenomenon and showed the necessity of using a "post-hoc correction" to maintain the fairness guarantee despite privatization (Mozannar et al., 2020). Similarly, a line of work has emerged to quantify and mitigate the fairness cost of employing privacy preserving practices (Tran et al., 2021a; 2022a). All PATE designs where fairness intervention occurs before the voting mechanism (the noisy argmax, which PATE derives its privacy guarantees from), namely ❶, ❷, and ❸ are (see Figure 1a), subject to this DP-induced fairness violations.

**(C2) Fairness pre-processing increases the cost of private training.** In Theorem 1, we show that pre-processing the training data to equalize subpopulation rates will degrade the privacy guarantee of the composed mechanism (i.e., preprocessor followed by private learning) compared to that of the private learning mechanism alone :

**Theorem 1.** *Assume the training dataset $D = \{(\mathbf{x}, z, y) \mid \mathbf{x} \in \mathcal{X}, z \in \mathcal{Z}, y \in \mathcal{Y}\}$ is fed through the demographic parity pre-processor $\mathcal{P}_{pre}$ following an ordering defined over the input space $\mathcal{X}$. Let $\mathcal{P}_{pre}$ enforce a maximum violation $\gamma$, and $|Z| = 2$. Suppose now $\mathcal{M}$ is an $(\varepsilon, \delta)$ training mechanism, then $M \circ \mathcal{P}_{pre}$ is $(K_\gamma \varepsilon, K_\gamma e^{K_\gamma \varepsilon} \delta)$-DP where $K_\gamma = 2 + \left\lceil \frac{2\gamma}{1-\gamma} \right\rceil$.*

We provide the proof for Theorem 1 in Appendix A. This result is training-framework-agnostic, and highlights the inefficacy of pre-processing in a privacy-aware learning pipeline. In Figure 1a, ❶ can be regarded as fairness pre-processors, and as such, suffers from a deteriorated privacy guarantee.

**(C3) Inefficient Privacy Budgeting.** In PATE, integrating unfairness mitigation after the privacy intervention can lead to inefficient privacy budgeting. This is because due to PATE's design, rejecting a query always incurs smaller costs than answering it. If now, queries are answered without rejection due to privacy constraint, but later on filtered out to mitigate unfairness, this privacy budget is wasted. In reality, no privacy is actually lost, but the privacy budgeting has been made inaccurate (the bound between actual and reported privacy becomes looser) which leads to unnecessary utility loss. In Figure 1a, all designs that apply a fairness mechanism post privacy-noising, namely ❺, ❻, and ❼; suffer from this inefficient privacy budgeting.

For designs ❻ and ❼ we provide an instructive example in Figure 1b. Assume the student model is a SVM and we require demographic parity. The correct hypothesis class (the max margin solution) $h$ is in black. Assume that the outlier query $x_0$ is privacy-costly such that $\varepsilon(x_0) \approx \varepsilon(x_1) + \varepsilon(x_2)$, but $x_0$ is the last query answered as the privacy budget is exhausted. Clearly no hypothesis $h'$ can achieve perfect accuracy or exact fairness. We show one such $h'$ in magenta. However, if query $x_0$ was not in the training set to begin with, we could have achieved both. Our design, FairPATE ❹, rejects $x_0$ due to fairness violation which frees up the budget to accept $x_1$ and $x_2$ and learn the more robust $h$.

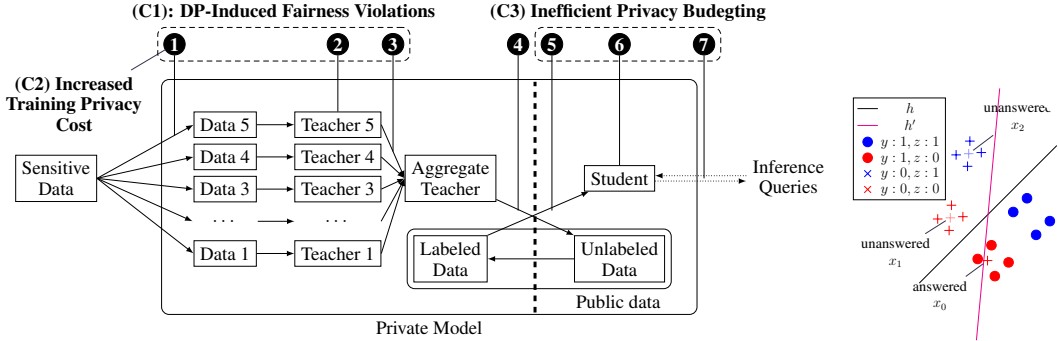

(a) **Various ways to integrate fairness in PATE.** For teachers: Pre- ❶/In- ❷/Post-Processing ❸. For the student: Pre- ❺/In- ❻/Post-Processing ❼. Dashed line separates public and private data domains. Our FairPATE's intervention occurs at ❹.

(b) **Inefficient Privacy Budgeting can lead to inaccurate and/or unfair models.**

## 4 FAIRPATE

In Section 3 we showed that the only viable fairness intervention is at ❹. We now present our design at this intervention point which allows us to apply the fairness intervention after the noisy argmax (addressing **C1**), and provide an accurate privacy accounting (addressing **C3**). Finally, at ❹, we are not making any additional passes on teacher data, therefore, also **C2** does not apply.

### 4.1 CONFIDENT&FAIR GNMAX: A PRIVACY-PRESERVING AND FAIR AGGREGATION ALGORITHM

As discussed in Section 2, the privacy of *standard PATE* is introduced at the level of teacher aggregation during the labeling of the public dataset where queries with public data are either rejected if they incur too high privacy costs, or labeled otherwise. We present FairPATE which introduces an unfairness mitigation to extend this private aggregation. Therefore, we propose a new aggregation mechanism, namely Confident&Fair-GNMax (**CF-GNMax**, see Algorithm 1) that extends PATE's standard GNMax algorithm (Algorithm 4 in Appendix G.2) with the idea of rejecting queries also due to their disparate impact on fairness.

Concretely, CF-GNMax, integrates an additional demographic parity constraint within the aggregator which allows rejecting queries on the basis of fairness. The algorithm checks potential demographic disparity violations and maintains an upper bound $\gamma$ on them in the course of answering PATE queries (Line 7 in Algorithm 1). The goal is to bound the actual $\Gamma(z, k)$—here empirically estimated. Concretely, we measure demographic disparity $\widehat{\Gamma}(z, k)$ using the counter $m : \mathcal{Z} \times \mathcal{K} \mapsto \mathbb{Z}_{\geq 0}$ which tracks per-class, per-subgroup decisions.

Care must be taken to produce accurate $\Gamma(z, k)$ estimations: with few samples, $\widehat{\Gamma}(z, k)$ may be a poor estimator of $\Gamma(z, k)$. Therefore, we have a cold-start stage where there are not yet enough samples to estimate $\widehat{\Gamma}(z)$ accurately. We avoid rejecting queries due to the fairness constraint at this stage.

---

**Algorithm 1 – Confident&Fair-GNMax Aggregator**

**Input:** query data point $x$, sensitive attribute $z$, predicted class label $k$, subpopulation subclass counts $m : \mathcal{Z} \times \mathcal{K} \mapsto \mathbb{Z}_{\geq 0}$

**Require:** minimum count $M$, threshold $T$, noise parameters $\sigma_1$, $\sigma_2$, fairness violation margin $\gamma$

1: **if** $\max_j \{n_j(x)\} + \mathcal{N}(0, \sigma_1^2) \geq T$ **then**
2:      $k \leftarrow \arg\max_j \{n_j(x) + \mathcal{N}(0, \sigma_2^2)\}$
3:      **if** $\sum_{\tilde{k}} m(z, \tilde{k}) < M$ **then**
4:          $m(z, k) \leftarrow m(z, k) + 1$
5:          **return** $k$
6:      **else**
7:          **if** $\left( \frac{m(z,k)+1}{\left(\sum_{\tilde{k}} m(z,\tilde{k})\right)+1} - \frac{\sum_{\tilde{z} \neq z} m(\tilde{z},k)}{\sum_{\tilde{z} \neq z, \tilde{k}} m(\tilde{z},\tilde{k})} \right) < \gamma$
       **then**
8:             $m(z, k) \leftarrow m(z, k) + 1$
9:             **return** $k$
10:      **else**
11:          **return** $\perp$
12: **else**
13:      **return** $\perp$

---

Concretely, we require at least, on average, $M$ samples from the query's subgroup before we reject a query on the basis of fairness (Line 3).

We note that *the unfairness mitigation in FairPATE occurs almost exactly at the same point as the privacy-utility balancing mechanism*. This allows us to get as close as possible to the simultaneous optimization of objectives which avoids the inefficient privacy budgeting **(C1)** discussed in Section 3.

**Privacy Analysis.** FairPATE's query phase (CF-GNMAX, Algorithm 1) has two main differences to PATE's (C-GNMAX, Algorithm 4). First, FairPATE involves a *cold-start* stage during which fairness violations estimators are inaccurate. During this stage, no fairness-related rejection takes place until all subgroups have at least $M$ samples. Second, in FairPATE, queries can be rejected for two reasons. Reason 1: Similar to standard PATE, queries that incur too high privacy costs are rejected. Reason 2: Additionally, queries whose answer would violate the fairness ($\gamma$-DemParity) constraint are rejected, as well. During the cold-start stage (Line 3), FairPATE follows the privacy analysis of PATE (Appendix B). Afterwards, queries can be rejected due to fairness. We can calculate FairPATE's overall probability of answering query $q_i$ as:

$$\mathbb{P}[\text{answering } q_i(z,k)] = \begin{cases} 0 & \text{if } \frac{m(z,k)+1}{(\sum_{\tilde{k}} m(z,\tilde{k}))+1} - \frac{\sum_{\tilde{z} \neq z} m(\tilde{z},k)}{\sum_{\tilde{z} \neq z, \tilde{k}} m(\tilde{z},\tilde{k})} > \gamma \\ \tilde{q} & \text{otherwise} \end{cases}$$

where $k$ is the noisy argmax (Line 2 in Algorithm 1), $\tilde{q}$ is calculated using Proposition 1 in Appendix B (as before), and the left side of the condition is simply calculating the new tentative demographic disparity violation $\Gamma(z,k)$ if the query is accepted. Note that in PATE (and by extension FairPATE) queries come from a public (and therefore non-private) dataset, and are labeled, noised and only then used to increment $m(z,k)$. Therefore, since the value of the counter $m(z,k)$ is only conditioned on the value of the noisy argmax, by the post-processing property of DP (Dwork & Roth, 2013), $m(z,k)$ and by extension, Line 7 do not add any additional privacy cost, *i.e.*, rejecting queries on the basis of fairness, does not incur additional privacy cost.

## 4.2 EXTENDING FAIRPATE TO OTHER FAIRNESS METRICS

Many group fairness metrics use not only the sensitive subgroup information $Z$ and prediction label (or score) $\hat{Y}$, but also ground truth labels $Y$. These include Equality of Odds (EO) (Hardt et al., 2016), Error (loss) Disparity (Bagdasaryan et al., 2019) and Equality of False Positives (Zafar et al., 2019). This is not an issue for unfairness mitigations that occur at training time (aka, in-processing). However, at inference time we do not have access to $Y$. Since PATE, and FairPATE by extension, are inference-time algorithms, the noisy mechanism which aggregates teacher votes does not have access to $Y$ and therefore cannot reject queries due to a violation of, for example, EO.

We propose to remedy this issue by using a calibration set $D_C$ which has ground-truth $Y$ labels. This dataset can be a small labeled subset of the public data. Next, we re-formulate a given group fairness metric's probability terms into ones that involve only $\hat{Y}, Z$, and the rest of the terms that involve $(\hat{Y}, Y, Z)$. We then estimate the latter terms on the calibration set $D_C$. We provide the theoretical decomposition for EO next and differ decomposition of other metrics to Appendix C.

**Decomposing Equality of Odds (EO).** The fairness metric for EO is $\Gamma_{\text{EO}}(Y, \hat{Y}, z) = \max_{k' \in \mathcal{Y}} \Pr[\hat{Y} = k' \mid Y = k, Z = z] - \Pr[\hat{Y} = k' \mid Y = k, Z \neq z]$. We only decompose the first term (for $z = Z$) using law of total probabilities. The second term is similar.

$$\Pr[\hat{Y} = k' \mid Y = k, Z = z] \approx \frac{P_D[\hat{Y} = k' \mid Z = z] - P_{D_C}[\hat{Y} \neq k' \mid Y \neq k, Z = z] P_D[\hat{Y} \neq k']}{P_D[\hat{Y} = k']}$$

$$= \frac{P_D[\hat{Y} = k' \mid Z = z]}{P_D[\hat{Y} = k']} - \frac{P_D[\hat{Y} \neq k']}{P_D[\hat{Y} = k']} P_{D_C}[\hat{Y} \neq k' \mid Y \neq k, Z = z]$$

where $P_D$ and $P_{D_C}$ are empirical probability estimations on the query set $D$ and the calibration set $D_C$. We provide a similar decomposition for loss disparity in Appendix C. In practice, the $P_{D_C}$ term(s) are constant *calibration* factors in the form of ratios of counts that estimate various marginal probabilities over $(Y, \hat{Y}, Z)$. We calculate them offline once for every dataset before any query is actually answered.

We make a note that a public labeled dataset can also be used to fine-tune a trained model with a fairness regularized training loss. If used on its own, as a post-DP intervention, fine-tuning is also

subject to inaccurate privacy budgeting (see **C3** in Section 3); and therefore, it is not a replacement for the calibration technique we discussed in this section. However, the two can be used in conjunction, to achieve both an accurate privacy accounting and tighter fairness guarantees for the model. We explore this in Appendix D.

## 4.3 ENSURING EXACT FAIRNESS AT INFERENCE-TIME VIA A REJECT OPTION

In this section, we extend FairPATE (independent of the used metric) within a larger decision making context where no unfair algorithmic decisions are tolerated. In this case, the model is allowed to refuse to answer rather than produce an unfair decision.

To guarantee and enforce the required degree of fairness at inference-time, we propose an inference-time post-processor (IPP) highlighted in Algorithm 2. Our design is inspired by our fairness mitigation in FairPATE, but is model-agnostic and generally applicable. At its core, IPP adds a reject option to the classification (often referred to as *selective classification* (Geifman & El-Yaniv)). In selective classification the trade-off is between accuracy and coverage (the percentage of queries that are answered). However, our use of the reject option trades off coverage with fairness at inference-time. That is we reject queries to ensure a bound on fairness violations. At inference time, our IPP keeps track of the counts of positive predictions per subpopulations. For every query posed to the model, it first calculates the demographic disparity based on the current counters. Then, it returns a label only if the resulting success rate of the current sub-population (in comparison to the other sub-populations) stays within the tolerated fairness violation. Since, at the beginning of inference, the model has not returned enough predictions to reliably estimate the

---

**Algorithm 2 Inference-time Post-Processor (IPP)**

---

**Input:** data point $x$, sensitive attribute $z$, predicted label $\hat{y}$, subpopulation-class counts $m : \mathcal{Z} \times \mathcal{Y} \mapsto \mathbb{Z}_{\geq 0}$
**Require:** minimum count $M$, fairness violation margin $\gamma$
1: **if** $\sum_{\tilde{y}} m(z, \tilde{y}) < M$ **then**
2:      $m(z, y) \leftarrow m(z, \hat{y}) + 1$
3:      **return** $\hat{y}$
4: **else**
5:      **if**
$$\left( \frac{m(z, \hat{y}) + 1}{(\sum_{\tilde{y}} m(z, \tilde{y})) + 1} - \frac{\sum_{\tilde{z} \neq z} m(\tilde{z}, \hat{y})}{\sum_{\tilde{z} \neq z, \tilde{y}} m(\tilde{z}, \tilde{y})} \right) < \gamma$$
     **then**
6:          $m(z, y) \leftarrow m(z, \hat{y}) + 1$
7:          **return** $\hat{y}$
8:      **else**
9:          **return** $\perp$

---

per-sub-population success rates, we propose a cold-start phase (line 1-3), similar to in FairPATE, during which all queries are answered.

## 5 EMPIRICAL EVALUATION

In our evaluation of FairPATE, we seek to **(a)** compare our solution to the state-of-the-art in training fair and private ML models; **(b)** validate the claims in Section 3, and subsequently the design choice for FairPATE; and **(c)** evaluate FairPATE in a classification setup with reject option (see Section 4.3).

**Experimental Setup.** We evaluate FairPATE on tabular fairness and privacy benchmark datasets from Lowy et al. (2023), namely, Adult (Becker & Kohavi, 1996), Retired Adult (Ding et al., 2022), German Credit Card (Hofmann, 1994), and Parkinson's (Little, 2008), as well as the vision dataset UTKFace (Zhang et al., 2017). The classification task for Credit-Card and Parkinson's is "whether the user will default payment the next month", and "whether the total UPDRS score of the patient is greater than the median or not," respectively. For Adult and Retired Adult, the task is "whether the individual will make more than $50K." In all tasks, the sensitive attribute is gender.

For the tabular data, we adopt the experimental setup of Lowy et al. (2023) (including classification task, pre-processing, etc.) and report the baselines results directly from the official repository (Gupta, 2023). For UTKFace, we evaluate DP-FERMI (Lowy et al., 2023) on gender classification task using the same architecture (ResNet-50) and pre-training on ImageNet, as we do for FairPATE and our other baselines. See Table 1 in Appendix F for more details on the experimental setup. All tabular baselines use logistic regression for student and teacher models.

**SOTA Baseline Comparisons.** We evaluate FairPATE against DP-FERMI (Lowy et al., 2023) (the current state-of-the-art) and also include their reported results for Tran et al. (2021a) and Jagielski et al. (2019). We show our results on the Credit-Card and Parkinson's datasets in Figure 2 under the Demographic Parity (DemParity), and in Figure 3 for Equality of Odds (EqOdds). As an overall trend, we observe that FairPATE outperforms the baselines in terms of fairness-accuracy trade-offs for most

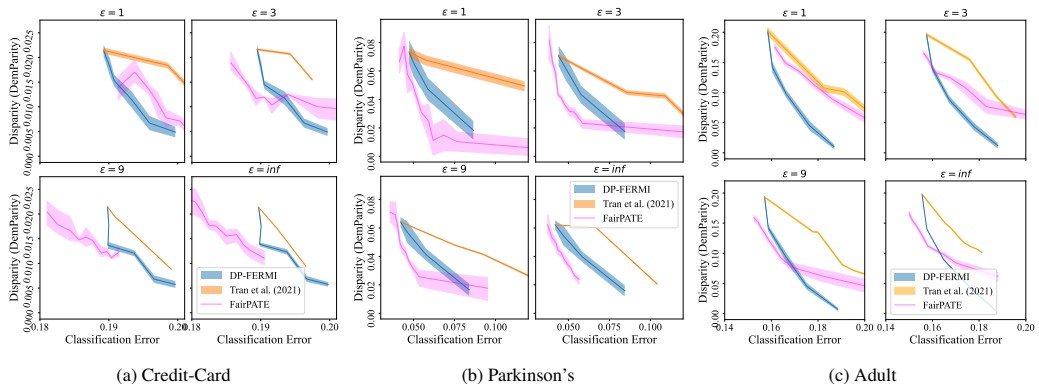

Figure 2: **Demographic Parity on Credit-Card, Parkinson's and Adult datasets.** DP-FERMI produces better trade-offs at smaller $\varepsilon$ levels, but FairPATE does better for larger $\varepsilon$'s. On average, FairPATE matches DP-FERMI with very similar disparity and classification errors (within 1%)

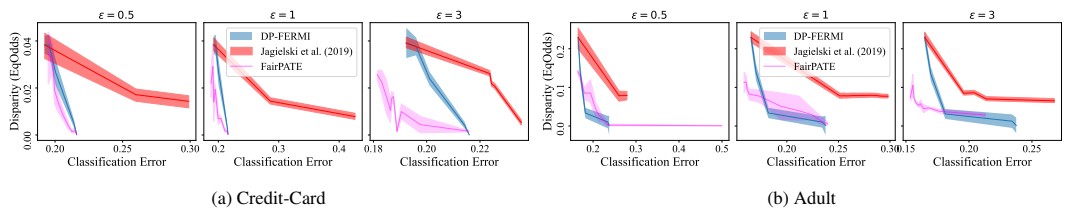

Figure 3: **Equality of Odds on Credit-Card and Adult datasets.** FairPATE consistently produces better fairness-accuracy trade-offs. On average, FairPATE achieves 2% lower disparity with slightly lower error.

privacy budgets on these datasets. In particular, under EqOdds, FairPATE consistently produces better fairness-accuracy trade-offs. On average, FairPATE achieves on average 2% lower disparity with a slightly lower error. Under DempParity, DP-FERMI produces better trade-offs at smaller $\varepsilon$ levels, but FairPATE does better for larger $\varepsilon$'s (see Figure 2c). On average, however, FairPATE matches DP-FERMI with very similar disparity and classification errors (within 1%). We present further experimental results in Appendix E. In Figure 6 we repeat our comparisons for a Imagenet pre-trained resnet-50 model trained and evaluated on UTKFace under Demographic Parity; and observe similar outcomes.

The previous comparisons highlight a key difference between DP-FERMI and our FairPATE. DP-FERMI extends further to lower disparity regions compared to FairPATE. This is because DP-FERMI employs a regularization technique which operates in the model's weight space and can enforce the fairness constraint with a large penalty. However, this comes at the expense of generalization performance (Figure 2a). FairPATE, on the other hand, operates in the sample space where the fairness intervention is limited to rejecting samples that violate the fairness constraint. Depending on the data distribution, this lower bounds the degree to which the fairness gap can be reduced. For instance, in Parkinson's (Figure 2b) which is a label-balanced dataset, FairPATE matches or improves on DP-FERMI for most $\varepsilon$'s, whereas in Adult (with a 1:3 ratio of labels) this is not the case. As additional ablation, in Appendix D, we present a weight space intervention on a trained FairPATE model that allows us to further enforce the fairness constraint with a moderate cost to generalization.

**FairPATE Design Validation.** We empirically validate claims in Section 3 by comparing FairPATE to two other PATE-based baselines which use the standard PATE's query selection process. **PATE-S$_{pre}$** (❺ in Figure 1a) incorporates an unfairness pre-processor for the Student model. This pre-filters the training data points on which the student will be trained on (see Algorithm 3 for a full description of the pre-processor). **PATE-S$_{in}$** ❻, has a fairness-regularized training loss (an in-processor for the Student). This implies training the student on all the queries labeled by the teachers, but setting additional constraints during training. For tabular data, we use fairlearn implementation of Exponentiated Gradient (Agarwal et al., 2018; Weerts et al., 2023). Both baselines serve to

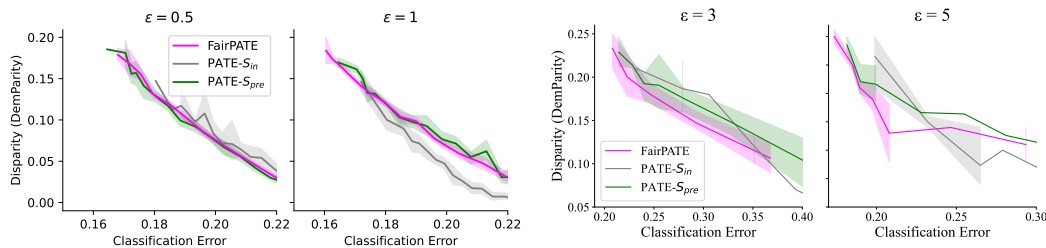

Figure 4: **Baseline Comparisons: Demographic Parity on Adult (left) and UTKFace (right) datasets.** While the performance of FairPATE and the two baselines are similar, FairPATE generally achieves better accuracy at the same fairness and privacy levels.

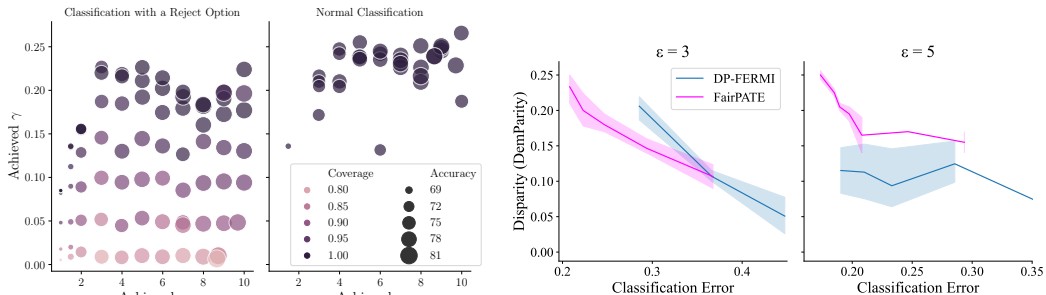

Figure 5: **Reject Option for Fairness.** Inference-time query rejections helps achieve small fairness violations by decreasing coverage. It preserves high model accuracy in low fairness violation settings.

Figure 6: **DP-FERMI Comparisons: Demographic Parity on UTKFace dataset.** At low privacy budget, FairPATE produces higher accuracy at the same fairness level. At higher privacy budget, DP-FERMI has better trade-offs.

understand the impact of implementing fairness *after* the noisy aggregation of teachers, *i.e.*, a post-DP intervention.

Figure 4 compares FairPATE to the two baselines on Adult and UTKFace datasets. On both datasets, the PATE-based models have similar behavior, particularly at low privacy budgets. PATE-$S_{in}$ exhibits a similar pattern to DP-FERMI as both use in-processor operating in the weight space: tighter disparity control (up to 3% at $\varepsilon = 1$ on Adult) at the expense of lower generalization performance (up to 2%). PATE-$S_{pre}$ is the closest PATE-based baseline to FairPATE, with the only difference being the privacy accounting. On Adult, we observe an slightly improved fairness-privacy trade-off as a result of the more accurate privacy accounting of FairPATE. The improvements are more stark on UTKFace, especially in low fairness violation regions (Figure 4; figures on the right). Overall, FairPATE outperforms PATE-$S_{pre}$ in most settings.

**Reject Option for Fairness.** We evaluate FairPATE under the setting where no fairness violations are tolerated at inference time. Rejecting queries at inference introduces "coverage" into the trade-offs. Coverage indicates the percentage of classification requests answered. We showcase the resulting 4-way Pareto frontier of fairness, privacy, accuracy, and coverage in Figure 5. As expected, fairness no longer has a significantly impact on accuracy, but it is now in trade-offs with coverage instead, with lower fairness violations corresponding to lower coverages. Interestingly, we observe a substantially different privacy-accuracy trade-off under the reject option. Notably, a decrease in privacy budget (right to left, almost constant size) does not show a trade-off with accuracy until very low ($\varepsilon < 2$) privacy budgets. That is, *under the reject option, privacy, fairness and accuracy need not be in trade-off.* We attribute the sudden decrease in accuracy below $\varepsilon = 2$ to the fact that very few queries get answered, and that therefore the student model is not well-generalized.

## 6 Related Work

In terms of privacy notion, the majority of the prior work considers DP that only protects privacy of individuals with respect to particular sensitive features (which we denote by *DP-S* from hereon) (Jagielski et al., 2019; Mozannar et al., 2020). DP-S is a different privacy notion than Definition 2. DP-S does not protect the privacy of other (presumed non-sensitive) features. Since these features are not considered during the privatization process, they can still lead to catastrophic privacy failures (Dwork & Roth, 2013) that Definition 2 protects against.

While Tran et al. (2022b) implement a demographic parity intervention in PATE Student and Teacher models (❷, and ❻ in Figure 1a), they still operate under a DP-S privacy notion. FairPATE in contrast, is a general framework for central DP training, compatible with any group fairness metric.

To the best of our knowledge, the only other work that provides a central DP guarantee together with a group fairness intervention is Lowy et al. (2023). They propose a DP-SGD-based method with fairness-specific privacy accounting which initially considers a DP-S setting but extends it to central DP. Given the proximity to our work, we rely on Lowy et al. (2023) as a main basis for comparison. Zhang et al. (2021) also proposed a DP-SGD-based method but do not feature a fairness-specific privacy analysis. Instead, they rely on typical DP-SGD privacy accounting and early stopping criteria to reduce the number of training steps and, as a consequence, the privacy cost.

In the intersection between privacy and fairness, there is work that considers the fairness impact of DP learning itself (Esipova et al., 2022; Kulynych et al., 2022; Tran et al., 2021a; 2022a; Suriyakumar et al., 2021; Farrand et al., 2020). In particular, several works have considered and provided mitigations for the disparate impact on accuracy of DP-SGD, under a loss parity notion of fairness. For instance, Kulynych et al. (2022) present DP-IS-SGD which adds an importance sampling step to DP-SGD training with Poisson sampling and RDP accounting . There is a debate whether DP learning presents a trade-off with loss parity fairness (Berrada et al., 2023). Indeed, in Appendix E.3, we show that for FairPATE, error (loss) parity results are least affected by the introduction of differential privacy training compared to demographic parity and equality of odds.

## 7 Discussion

We presented FairPATE, the first PATE-based framework that provides (central) DP (*i.e.*, privacy guarantees for all features) with a group fairness intervention. Given the two-model architecture of PATE, we analyzed several possible fairness intervention points. We found out that placing the intervention with respect to the *training process* of either model—as is common in algorithmic fairness literature—leads to sub-optimal privacy-fairness-accuracy trade-offs. Our FairPATE intervention occurs in between, as close as possible to the privacy intervention which affords us an accurate privacy accounting.

As a result of the accurate privacy accounting, FairPATE is competitive with the state-of-the-art DP-SGD-based solution (DP-FERMI). However, the two methods have a key difference. In DP-FERMI both the fairness and privacy intervention occur in the weight space (*i.e.*, space of model parameters) while in FairPATE, both interventions occur in the sample space. From an algorithmic fairness perspective, this difference is akin to a pre-processors and in-processor.[1] Both have their stenghts and shortcomings. For instance, since in-processors allow for a larger degree of freedom to reduce the fairness gap (as we have also observed empirically). However, in-processing comes at the cost of stability (Huang & Vishnoi, 2019; Friedler et al., 2018). Being in the sample space, FairPATE inherits the advantages of PATE and is, for instance, architecture-agnostic and amenable to distributed learning under confidentiality (secrecy) à la Choquette-Choo et al. (2021).

Finally, regarding the rejecti-option for fairness, we note that in a human-in-the-loop system, the mistakes of the human would ultimately be mistakes of the end-to-end system; and therefore, measuring the efficacy of the system should consider the human error as well. We have shown that FairPATE and IPP enable such applications but whether such a system is the appropriate choice for a given application is out of the scope of the current study.

---

[1]From the student model point of view, FairPATE is a pre-processor

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

## A  PRIVACY COST OF PRE-PROCESSING

---

**Algorithm 3 Pre-Processor $\mathcal{P}_{\text{pre}}$**

---

**Input:** data point $x$, sensitive attribute $z$, true label $y$, subpopulation-class counts $m : \mathcal{Z} \times \mathcal{Y} \mapsto \mathbb{Z}_{\geq 0}$
**Require:** minimum count $M$, fairness violation margin $\gamma$
1: **if** $\sum_{\tilde{y}} m(z, \tilde{y}) < M$ **then**
2:     $m(z, y) \leftarrow m(z, y) + 1$
3:     **return** $x$
4: **else**
5:     **if** $\left( \frac{m(z, \hat{y}) + 1}{(\sum_{\tilde{y}} m(z, \tilde{y})) + 1} - \frac{\sum_{\tilde{z} \neq z} m(\tilde{z}, \hat{y})}{\sum_{\tilde{z} \neq z, \tilde{y}} m(\tilde{z}, \tilde{y})} \right) < \gamma$ **then**
6:         $m(z, y) \leftarrow m(z, y) + 1$
7:         **return** $x$
8:     **else**
9:         **return** $\perp$

---

Fairness pre-processing can lead to increased privacy costs during private training. A consequence of differential privacy is the privacy consumption regime Mironov (2017): just by observing the data for the purposes of equalizing a fairness measure between subpopulations, we may consume from the privacy budget.[2] This budget could otherwise be spent, for instance, on more training passes on data to yield higher accuracy. We formalize this observation in Theorem 1 for the case when a universal ordering exists.

*Proof.* We will proceed to show that using a pre-processing that sorts through data following some ordering defined over the whole input space [3] and, for any given label $y$, removes the last datapoints (following the ordering) in the majority subclass until it satisfies the $\gamma$-constraint will produce datasets at most $2 + K_\gamma = 2 + \left\lceil \frac{2\gamma}{1-\gamma} \right\rceil$ apart. One then applies group privacy to obtain the final claim of the theorem.

Let $D' = D \cup x^*$, and the label of $x^*$ is $y^*$. We now proceed to analyze how far apart $\mathcal{P}_{\text{pre}}(D)$ and $\mathcal{P}_{\text{pre}}(D')$ can be. First note, they are the same on all labels not $y^*$, so we need only consider the difference on this label. First, let $m$ be the size of the minority subclass for label $y^*$ and let $m + c$ be the admissible size of the majority class. That is, we have $\frac{m}{2m+c} - \frac{m+c}{2m+c} < \gamma$. From this we can conclude $c = \lfloor \frac{\gamma}{1-\gamma} 2m \rfloor$. Given this relation between the size of majority class a function of the minority class, we proceed to go through all logical cases to show the maximum difference is as claimed above.

Suppose $x^*$ belongs to the minority subclass for $y^*$ in $D$. Then we have $m \to m + 1$ and hence $c \to \lfloor \frac{\gamma}{1-\gamma} 2(m + 1) \rfloor$. Thus we see $\mathcal{P}_{\text{pre}}(D')$ now admits one more point in the minority class of $y^*$ and at most $1 + \lceil \frac{2\gamma}{1-\gamma} \rceil$ more points to the the majority subclass (note we do not replace existing points as we follow the ordering on the input space). Thus the max change between $\mathcal{P}_{\text{pre}}(D)$ and $\mathcal{P}_{\text{pre}}(D')$ is $2 + \lceil \frac{2\gamma}{1-\gamma} \rceil$

Now suppose $x^*$ belongs to majority subclass for $y^*$ in $D$. In this case we have either $x^*$ appears early enough in the ordering that it now replaces another point in the majority class when applying $P$, or it is not added. In the former case, this mean we have changed $\mathcal{P}_{\text{pre}}(D)$ by 2: we first removed a point and then added $x^*$. In the latter case, $x^*$ did not get added into the dataset, more so because of the ordering, $\mathcal{P}_{\text{pre}}(D') = \mathcal{P}_{\text{pre}}(D)$ as the order of points before $x^*$ is still the same. So in this case, once again, the change between $\mathcal{P}_{\text{pre}}(D)$ and $\mathcal{P}_{\text{pre}}(D')$ is less than $2 + \lceil \frac{2\gamma}{1-\gamma} \rceil$.

Thus we have by group privacy (see lemma 2.2 in Vadhan (2017)) that $\mathcal{M} \circ \mathcal{P}_{\text{pre}}$ gives the claimed DP-guarantee, as we set $K_\gamma = 2 + \lceil \frac{2\gamma}{1-\gamma} \rceil$

$\square$

---

[2]Note that this disadvantage does not hold for fairness post-processing which does not incur additional privacy costs due to the differential privacy post-processing property.

[3]An example of such ordering would be to order images based on their pixel values in some specified order of height, width and channel starting by checking the first pixel, then the second pixel, and so on.

## B  STANDARD PATE PRIVACY ANALYSIS

Papernot et al. Papernot et al. (2018) use Rényi differential privacy (RDP) Mironov (2017) for accounting of the privacy budget expanded in answering each query. While the true privacy cost for each query is not known, an upperbound is estimated and summed over the course of the query phase. Answering queries stop when a pre-defined budget is exhausted. A student model is then trained on the answered queries.

Theorem 2 establishes that the upperbound is a function of the probability of *not* answering a query $i$ with the plurality vote $i^*$. Unsurprisingly, this privacy cost function must tends to zero when the said event is very unlikely (*i.e.*, strong consensus):

**Theorem 2** (From Papernot et al. (2018)). *Let $\mathcal{M}$ be a randomized algorithm with $(\mu_1, \varepsilon_1) - RDP$ and $(\mu_2, \varepsilon_2) - RDP$ guarantees and suppose that given a dataset $D$, there exists a likely outcome $i^*$ such that $\Pr[\mathcal{M}(D) \neq i^*] \leq \tilde{q}$. Then the data-dependent Rényi differential privacy for $\mathcal{M}$ of order $\lambda \leq \mu_1, \mu_2$ at $D$ is bounded by a function of $\tilde{q}, \mu_1, \varepsilon_1, \mu_2, \varepsilon_2$, which approaches 0 as $\tilde{q} \to 0$.*

In practice, Proposition 1 is used to find $\tilde{q}_i$ in Theorem 2, and $\mu_1, \mu_2$ are optimized to achieve the lowest upperbound on the privacy cost of each query for every order $\lambda$ of RDP.

**Proposition 1** (From Papernot et al. (2018)). *For any $i^* \in [m]$, we have $\Pr[\mathcal{M}_\sigma(D) \neq i^*] \leq \frac{1}{2} \sum_{i \neq i^*} \mathrm{erfc}\left(\frac{n_{i^*} - n_i}{2\sigma}\right)$, where erfc is the complementary error function.*

## C  FAIRNESS METRIC DECOMPISITIONS

**Loss Disparity.** Consider Loss (Error) Disparity as our fairness notion:

$$\Gamma_{\mathrm{error}}(Y, \hat{Y}, z) = \Pr[Y \neq \hat{Y} \mid Z = z] - \Pr[Y \neq \hat{Y} \mid Z \neq z].$$

$$
\begin{aligned}
\Pr[Y \neq \hat{Y} \mid Z = z] &= \sum_k \Pr[Y = k, \hat{Y} \neq k \mid Z = z] \Pr[Y = k] \\
&= \sum_k \Pr[\hat{Y} \neq k \mid Z = z] \Pr[Y = k \mid \hat{Y} \neq k, Z = z] \Pr[Y = k] \\
&= \sum_k \sum_{k' \neq k} \underbrace{\Pr[\hat{Y} = k' \mid Z = z]}_{\text{estimate on unlabeled public data}} \underbrace{\Pr[Y = k \mid \hat{Y} = k', Z = z] \Pr[Y = k]}_{\text{estimate on labeled public data}} \\
&\approx \sum_k \sum_{k' \neq k} P_D[\hat{Y} = k' \mid Z = z] \, P_{D_C}[Y = k \mid \hat{Y} = k', Z = z] P_{D_C}[Y = k] \\
&= \sum_{k'} \sum_k P_D[\hat{Y} = k' \mid Z = z] \, P_{D_C}[Y = k \mid \hat{Y} = k', Z = z] P_{D_C}[Y = k] \mathbb{1}_{\{k' \neq k\}} \\
&= \sum_{k'} \underbrace{P_D[\hat{Y} = k' \mid Z = z]}_{A: |\mathcal{Y}| \times |\mathcal{Z}|} \underbrace{\sum_k \underbrace{P_{D_C}[Y = k \mid \hat{Y} = k', Z = z]}_{|\mathcal{Y}| \times |\mathcal{Z}| \times |\mathcal{Y}|} \underbrace{\mathbb{1}_{\{k' \neq k\}}}_{|\mathcal{Y}| \times |\mathcal{Y}|} \underbrace{P_{D_C}[Y = k]}_{|\mathcal{Y}| \times 1}}_{B: |\mathcal{Y}| \times |\mathcal{Z}|} \\
&= \mathbf{1}_{|\mathcal{Y}| \times 1}^\top \cdot (A \odot B)
\end{aligned}
$$

## D  MODEL FAIR-TUNING

Our goal is to further reduce the disparity gap that a FairPATE student model has without accumulating additional privacy cost. We start by making two key observations: **(a)** in private training, we care about the privacy of training data but not inference data, and **(b)** in providing fairness guarantees for models, we care about the fairness gaps at inference-time. Therefore, we present a fine-tuning mechanism that takes advantage of public data to *adjust the model post-training but pre-inference*. The effect is that no privacy budget is spent as data used to calibrate the model is public, and on the other hand, the model exhibits better fairness gaps as a result.

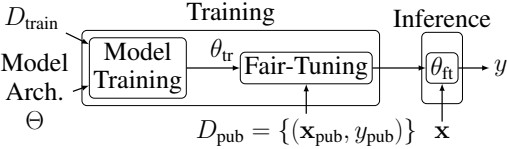

Figure 7: **Fair-Tuning using Public Data.** Note that model $\theta \in \Theta$ can be any trained model. Here we assume a FairPATE student.

**Method.** In Section 4.2, we introduced a calibration method where we used a *labeled* public dataset to estimate probability factors over ground truth labels offline. This enabled us to apply ground-truth-based fairness losses for PATE query answering. We required a relatively small dataset since the estimates were only for 3-dimensions $(\hat{Y}, Y, Z)$. If we have access to a larger sample of labeled queries, we can *fair-tune* our models. Figure 7 depicts our approach. Given any trained (student) model $\theta_{\text{tr}}$ as our initialization, we perform additional training steps using the following loss on a public dataset $D_{\text{pub}} = \{(\boldsymbol{x}, z, y)\}$:

$$\ell_{\text{fair-tune}}(\theta; D_{\text{pub}}) = \frac{1}{|D_{\text{pub}}|} \sum_{i \in |D_{\text{pub}}|} \ell_{\text{train}}(\theta(\boldsymbol{x}_i), y_i) + \lambda \max\{0, \Gamma(\theta(\boldsymbol{x}_i), y_i, z_i) - \gamma\}, \qquad (1)$$

where $\ell_{\text{train}}$ is the original training loss used for the student model. $\lambda$ is a hyper-parameter that ensures the balance between fine-tuned model accuracy and fairness.

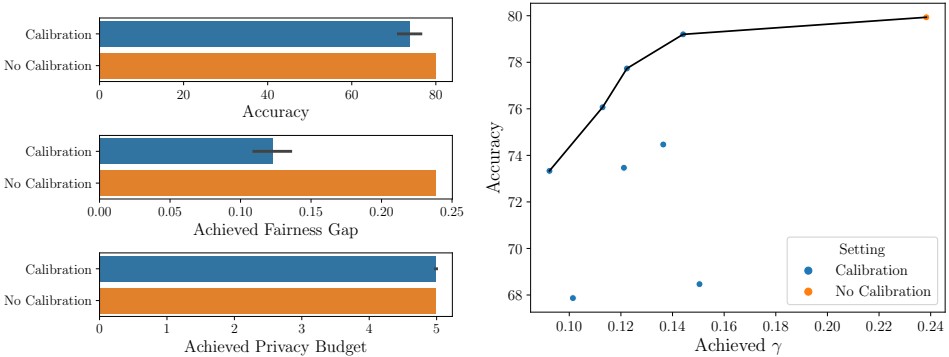

Figure 8: **Model Fair-Tuning trade-off fairness for accuracy without impacting the privacy budget**

**Empirical Results.** Our empirical result in Figure 8 on UTKFace using FairPATE without IPP, shows that fair-tuning helps reduce the fairness gap by about 9% (from 24% to 15%) with less than 2% drop in accuracy, for the model on the Pareto frontier shown in Figure 8 (right).

Our more extensive Pareto frontier in Figure 9 confirms our findings. In low privacy and low-fairness violation regions in particular, we see the largest gains from calibration. Calibration allows us to achieve better utility for the same privacy and fairness level of guarantees. In one case, for $\varepsilon = 1.48$ and $\gamma = 0.14$ we achieve a 9% improvement to accuracy (from 69% to 78%).

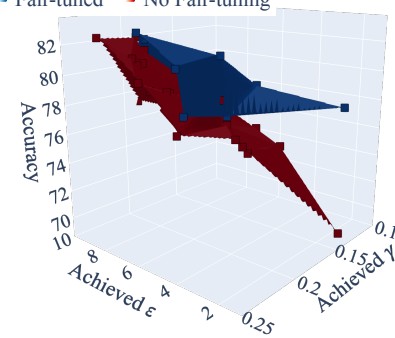

Figure 9: **FairPATE (w/o IPP) With (blue) and Without (red) model fair-tuning on UTKFace**

# E    ADDITIONAL EXPERIMENTAL RESULTS

In this section, we present additional experimental results.

## E.1 Baseline Comparisons with IPP

We present baseline comparisons with IPP in this section. We compare FairPATE to DP-SGD, PATE, PATE-S$_{pre}$, and PATE-S$_{in}$. We apply IPP to all methods for consistency.

Figure 10 shows sampled points from the Pareto frontier in 2d with size indicating accuracy and hue, coverage. We observe that at the same privacy budget and fairness violation, FairPATE usually has larger points with darker colours, indicating higher accuracy and higher coverage. This is shown clearly in low fairness violation regions at the bottom of the figures.

Figure 11 plots the Pareto frontier surface for each method. We plot achieved $\varepsilon$, achieved $\gamma$, and model accuracy on three axes. We show the model coverage with colours where lighter colours indicate lower coverage. The results show that FairPATE outperforms the baselines in terms of accuracy in high privacy budget low fairness violation regions. FairPATE also simultaneously achieves higher coverage than the baselines in these regions. Both DP-SGD and PATE do not have unfairness mitigation mechanisms, so it is not surprising that FairPATE outperforms them in regions with tight fairness constraints. FairPATE also performs better than PATE-S$_{pre}$ because it is able to save privacy budget when rejecting points that violate fairness constraint. It is then able to accept more data points to train a better student model. In low privacy budget regions and high fairness violation regions, FairPATE performs similarly to other methods because its fairness mechanism is not activated. The mechanism is not activated either due to having insufficient data point to move past the warm-up stage or fairness constraint being too loose.

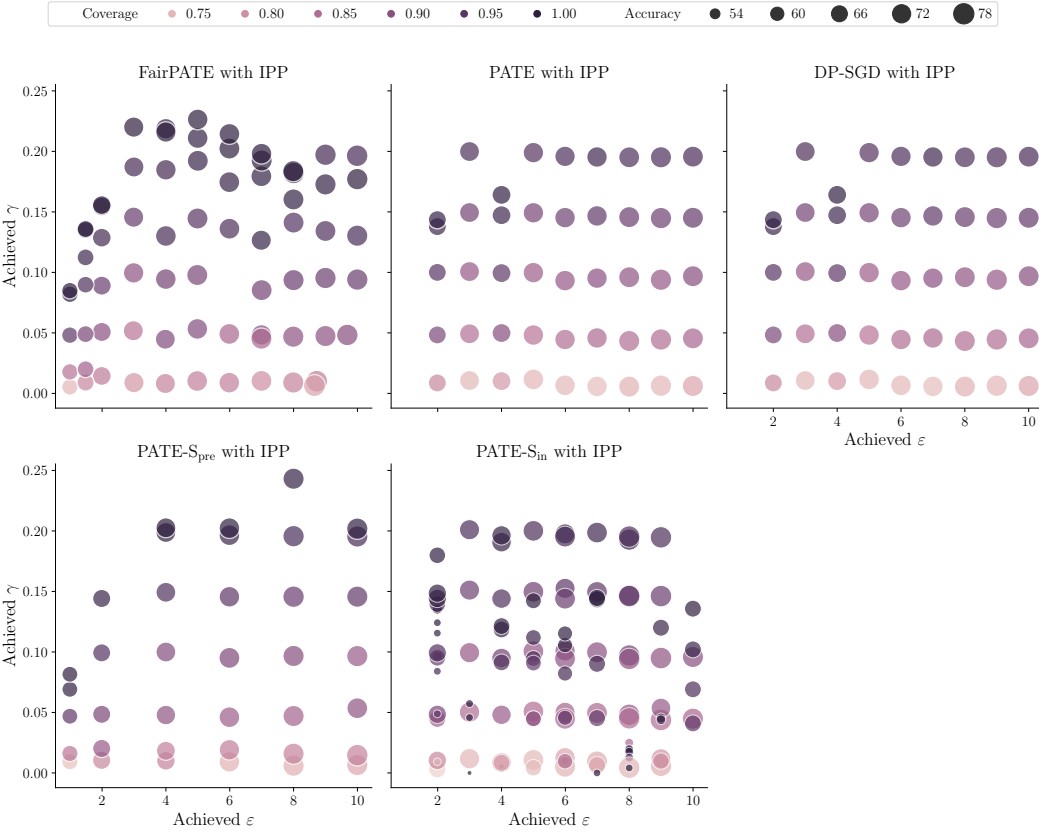

Figure 10: **FairPATE and baseline comparisons with IPP.** The size of the points indicates accuracy and the hue indicates coverage. FairPATE usually achieves higher accuracy and coverage than the baselines at the same privacy budget and fairness violation.

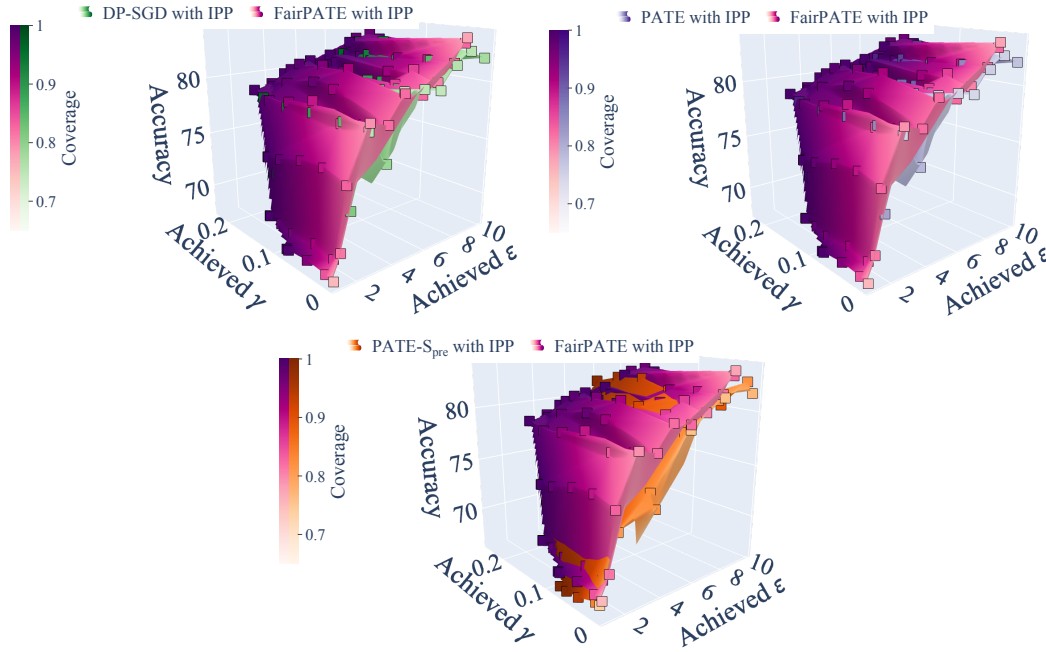

Figure 11: **FairPATE and baseline Pareto surface comparisons with IPP.** Colour of the surfaces shows model coverage. FairPATE outperforms baselines in most regions in both accuracy and coverage, especially in regions with high privacy budget and low fairness violations.

## E.2 Impact of Data Imbalance on IPP

In Figure 12 we show the fairness-accuracy trade-off curves of FairPATE under various privacy budgets; and with and without the presence of class imbalance. We show our results on Adult, since it is an imbalanced dataset with a 3:4 ratio of majority to minority class labels. In Figure 12b we balance the ratio 1:1 using `sklearn`'s `imbalanced-learn` undersampler. We are quick to note that error rates between Figure 12a and Figure 12b are not comparable (i.e., a random classifier achieves 25% error rate on the former and 50% on the latter).

Under class-imbalance for every percentage of coverage degradation, the fairness-privacy curve improves consistently (to lower left on the plot, with lower error and lower disparity). For the balanced case, we observe that the disparity levels are much higher. This is because random (non-stratified) sampling is not conducive to demographic parity. IPP's overall behavior is similar as before but the curves are much closer to each other and the rejection rates are higher due to high disparity levels of the initial models caused by random sampling.

## E.3 Comparisons between Different Group Fairness Metrics

In Figure 13, we show three FairPATE instances with different fairness notions, and compare their fairness-accuracy trade-offs. We observe that the model with error (loss) parity fairness is least affected by the introduction of the privacy intervention (DP) while the model with demographic parity intervention is most affected. Our observation regarding error parity is consistent with prior work (from Berrada et al. (2023), for instance) that has shown that for well-generalized models, differential privacy does not result in additional unfairness. We attribute this to the fact that error parity, is inherently a notion of generalization measured for sub-populations instead of over the whole dataset. A well-generalized model should in principle also generalize well for its marginal distributions as well.

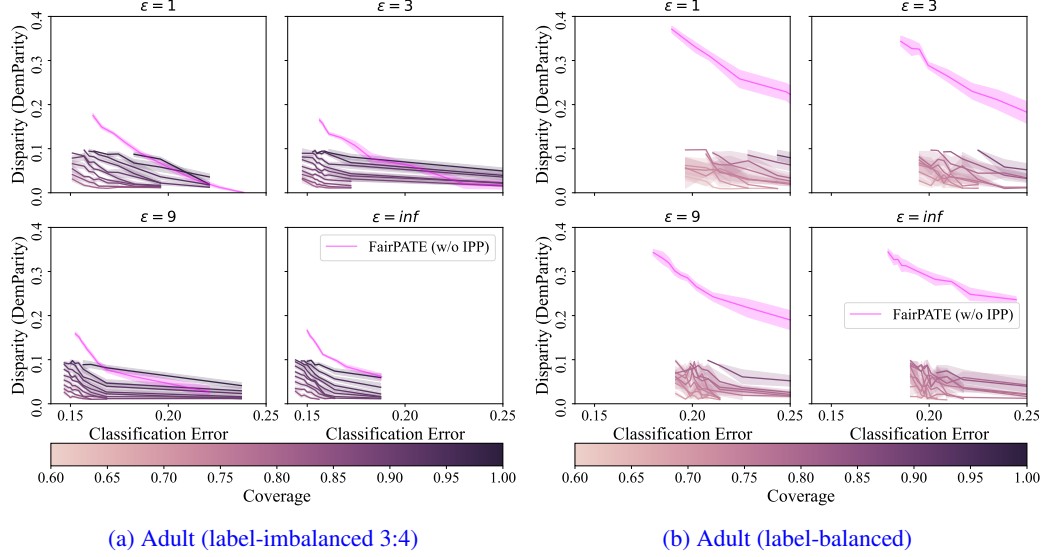

(a) Adult (label-imbalanced 3:4)   (b) Adult (label-balanced)

Figure 12: **IPP on Adult (label-imbalanced 3:4, left) vs. IPP on label-balanced Adult (right).** The colorbar marks the coverage (1-rejection rate) of the queries answered at inference time. Due to the balancing, the error rates are not comparable between (a) and (b).

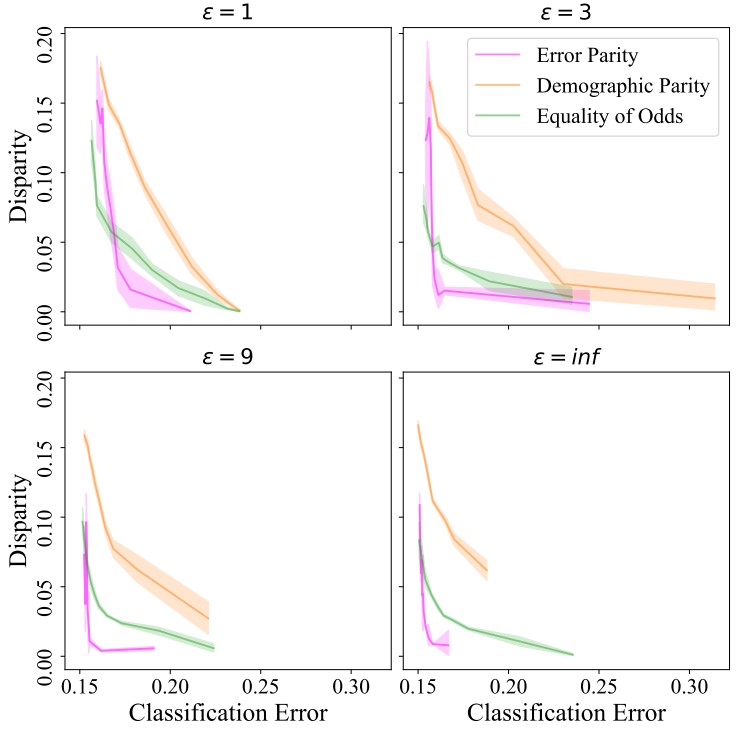

Figure 13: **Error Parity vs. Demographic Parity vs. Equality of Odds on Adult.** Models trained with loss parity notion of fairness are least affected by the introduction of differential privacy noising.

# F    EXPERIMENTAL SETUP

We split each dataset into a training set, an unlabeled set, and a test set. The sizes of these three datasets are determined based on the dataset sizes specified in original PATE Papernot et al. (2016;

2018), and adapted to the difficulties of the prediction tasks. For CheXpert, we only use the data from two races that have the most data. The other groups have too few data points for our fairness intervention to perform effectively.

In FairPATE, the training set is further split into equal partitions to train the teacher models. We train as many teachers as possible while still achieving good ensemble accuracy overall. In FairDP-SGD, the whole training set is used to train the private model. The test set is used to evaluate the performance of the final model.

## F.1 FAIRPATE

For FairPATE, we first train the teacher ensemble models, then query them with the public dataset, and aggregate their predictions using the FairPATE algorithm. The student model is trained on the public dataset with obtained labels. The model architectures, as well as the parameters used in querying the teacher models are detailed in Table 1 for each dataset, respectively. The model architectures are chosen by referencing what is used in related works for each dataset.

We tune the amount of noise injected into the aggregation mechanism of FairPATE by varying the standard deviation of the Gaussian distribution while ensuring the accuracy of the labels produced by the teacher ensemble models to maximize the accuracy of student models. We used a small validation set taken from the dataset to tune the FairPATE hyperparameters by training student models with different combinations of the hyperparameters and selecting the values that lead to the highest student model accuracy. The validation set is taken from the original training set and the size is half the size of the unlabeled set. When tuning, we vary the threshold T between 0.5 to 1.5 multiplied by the number of teacher models, $\sigma_1$ between 0 to the number of teacher models, and $\sigma_2$ between 0 to 0.5 multiplied by the number of teacher models.

We train the models using Adam optimizer. We use cross entropy loss function when training on ColorMNIST and binary cross entropy with logits on all the other datasets.

| Dataset | Prediction Task | C | Sens. Attr. | SG | Total | U | Model | Number of Teachers | $T$ | $\sigma_1$ | $\sigma_2$ |
|---|---|---|---|---|---|---|---|---|---|---|---|
| ColorMNIST (Arjovsky et al., 2019) | Digit | 10 | Color | 2 | 60 000 | 1 000 | Convolutional Network (Table 2) | 200 | 120 | 110 | 20 |
| CelebA (Liu et al., 2015) | Smiling | 2 | Gender | 2 | 202 599 | 9 000 | Convolutional Network (Table 3) | 150 | 130 | 110 | 10 |
| FairFace (Karkkainen & Joo, 2021) | Gender | 2 | Race | 7 | 97 698 | 5 000 | Pretrained ResNet50 | 50 | 30 | 30 | 10 |
| UTKFace (Zhang et al., 2017) | Gender | 2 | Race | 5 | 23 705 | 1 500 | Pretrained ResNet50 | 100 | 50 | 40 | 15 |
| CheXpert(Irvin et al., 2019) (Seyyed-Kalantari et al., 2020) | Disease | 2 | Race | 3 | 152 847 | 4 000 | Pretrained DenseNet121 | 50 | 30 | 20 | 10 |

Table 1: Datasets used for evaluation. Abbreviations: **C**: number of classes in the main task; **SG**: number of sensitive groups; **U**: number of unlabeled samples for the student training . Summary of parameters used in training and querying the teacher models for each dataset. The selection of $\sigma_1$ is in accordance with the threshold $T$. The selection process of $\sigma_2$, is shown in the Appendix F The pre-trained models are all pre-trained on ImageNet. We use the most recent versions from PyTorch.

| Layer | Description |
|---|---|
| Conv2D with ReLU | (3, 20, 5, 1) |
| Max Pooling | (2, 2) |
| Conv2D with ReLU | (20, 50, 5, 1) |
| MaxPool | (2, 2) |
| Fully Connected 1 | (4*4*50, 500) |
| Fully Connected 2 | (500, 10) |

Table 2: Convolutional network architecture used in ColorMNIST experiments.

| Layer | Description |
|---|---|
| Conv2D | (3, 64, 3, 1) |
| Max Pooling | (2, 2) |
| ReLUS | |
| Conv2D | (64, 128, 3, 1) |
| Max Pooling | (2, 2) |
| ReLUS | |
| Conv2D | (128, 256, 3, 1) |
| Max Pooling | (2, 2) |
| ReLUS | |
| Conv2D | (256, 512, 3, 1) |
| Max Pooling | (2, 2) |
| ReLUS | |
| Fully Connected 1 | (14 * 14 * 512, 1024) |
| Fully Connected 2 | (1024, 256) |
| Fully Connected 2 | (256, 2) |

Table 3: Convolutional network architecture used in CelebA experiments.

### F.2 Wall Time Measurements

We measure the wall time of running FairPATE compared to standard PATE. The setting we use and the results are shown in Table 4.

| Method | $\epsilon$ | $\gamma$ | Fairness Factor | Batch Size | Number of Epochs | Time |
|--------|-----|-----|-----------------|------------|------------------|------|
| FairPATE | 4 | 0.01 | N/A | 100 | 30 | 6min 14sec |
| PATE | 4 | N/A | N/A | 100 | 30 | 6min 43sec |

Table 4: Wall time measurements of different methods. All experiments use UTKFace dataset.

## G  Extended Background

In the following, we assume a classification task where a model $\theta : \mathcal{X} \times \mathcal{Z} \mapsto \mathcal{K}$ maps the features $(\mathbf{x}, z) \in \mathcal{X} \times \mathcal{Z}$ to a label $y \in \mathcal{K}$, where: $\mathcal{X}$ is the domain of non-sensitive attributes, $\mathcal{Z}$ is the domain of the sensitive attribute (as a categorical variable), and $\mathcal{K}$ is the domain of the output label (also categorical). Without loss of generality, we will assume $\mathcal{Z} = [Z]$ (*i.e.* $\mathcal{Z} = \{1, \ldots, Z\}$) and $\mathcal{K} = [K]$.

### G.1  Fairness Notion: Demographic Parity

We note that in a multi-class setting (*i.e.*, $K > 2$), and even in the binary-class settings where the problem does not admit a reasonable notion of the "desirable outcome", there can be multiple formulations of the notion of demographic parity. We adopt a natural extension of the well-known binary notion that requires equal rates for any class. Let us first define demographic disparity:

The *demographic disparity* $\Gamma(z, k)$ of subgroup $z$ for class $k$ is the difference between the probability of predicting class $k$ for the subgroup $z$ and the probability of the same event for any other subgroup: $\Gamma(z, k) := \mathbb{P}[\hat{Y} = k \mid Z = z] - \mathbb{P}[\hat{Y} = k \mid Z \neq z]$.

### G.2  Privacy Notion: Differential Privacy

In $(\varepsilon, \delta)$-DP, the parameter $\varepsilon$ bounds the maximal difference between the analysis results on the neighboring datasets while the second parameter $\delta$ represents a relaxation of the bound by allowing the results to vary more than the factor $e^\varepsilon$. Hence, the total privacy loss is bounded by $\varepsilon$ with a probability of at least $1 - \delta$ Dwork & Roth (2013). Note that smaller $\varepsilon$ correspond to better privacy guarantees for the data.

**PATE.** (Figure 1a), takes advantage of an unlabeled public data set $D_{\text{public}}$ to conserve the privacy of sensitive data $D_{\text{private}}$. Therefore, an ensemble of $B$ *teacher* models $\{\theta_i\}_{i=1}^B$ is trained using disjoint subsets of $D_{\text{private}}$ and their knowledge is transferred to a separate *student* model that can be publicly released. For the knowledge transfer, trained teachers label query data points from $D_{\text{public}}$. The final label of the query is a noisy argmax over the vote counts as $N(\mathbf{x}) = \arg\max \left([n_{i,j}]_{B \times K} + \mathcal{N}(0, \sigma_1^2)\right)$, where $K$ is the number of classes (see aggregation in Algorithm 4). Noising the argmax enables to implement the privacy guarantees according to DP.

PATE estimates the privacy cost of answering queries (*i.e.* labeling data) through *teachers consensus* with higher consensus revealing less information about individual teachers, and, thereby, consuming less privacy costs. To take advantage of the fact that estimating consensus is less privacy-costly than answering queries, PATE rejects high-cost queries to save on the privacy budget (see Algorithm 4). Both consensus estimation and vote aggregation (answering the query) are noised with $\mathcal{N}(0, \sigma_1^2)$ and $\mathcal{N}(0, \sigma_2^2)$, respectively; where $\sigma_1, \sigma_2$ are tuned for better student accuracy.

We include the standard Confident-GNMax Aggregator Algorithm from Papernot et al. (2018) below.

**Algorithm 4 – Confident-GNMax Aggregator (from Papernot et al. (2018))** given a query, consensus among teachers is first estimated in a privacy-preserving way to then only reveal confident teacher predictions.

---

**Require:** input $x$, threshold $T$, noise parameters $\sigma_1$ and $\sigma_2$
1: **if** $\max_j\{\sum_{i\in[B]} n_{i,j}(x)\} + \mathcal{N}(0,\sigma_1^2) \geq T$ **then**
2:     **return** $\arg\max_j\{\sum_{i\in[B]} n_{i,j}(\mathbf{x}) + \mathcal{N}(0,\sigma_2^2)\}$
3: **else**
4:     **return** $\perp$

---

**DP-SGD** extends standard stochastic gradient descent (SGD) with two additional steps to implement privacy guarantees. First, the individual data points' gradients are clipped to a maximum gradient norm bound $C$. This bounds the gradients' sensitivity, which ensures that no data points can incur changes to the model above magnitude $C$. After clipping, Gaussian noise with scale $\mathcal{N}(0,\sigma^2 C^2)$ is added to mini-batches of clipped gradients. The noise distribution has zero mean and standard deviation proportional to a pre-defined noise multiplier $\sigma$ and the clipping norm $C$. We detail the DP-SGD algorithm in Algorithm 5.

To yield tighter privacy bounds, DP-SGD implements a privacy amplification through subsampling Beimel et al. (2010): Training data points are sampled into mini-batches with a Poisson sampling per training iteration, in contrast to grouping the entire training data into mini-batches prior to every epoch as done in standard SGD. Hence, the traditional concept of an epoch (as a full training on the entire training data) does not exist in DP-SGD. Instead, each data point is sampled in every iteration according to a given sampling probability. Privacy amplification through subsampling allows to scale down the noise $\sigma$ by the factor $L/N$ (with $L$ being the expected mini-batch size, $N$ the total number of data points, and $L \ll N$) while still ensuring the same $\varepsilon$ as with $\sigma$ Kairouz et al. (2021) which is crucial to the practical performance (privacy-utility trade-offs) of DP-SGD.

We include the standard DP-SGD algorithm (Algorithm 5):

---

**Algorithm 5** Standard DP-SGD, adapted from Abadi et al. (2016).

---

**Require:** Private training set $D_{\text{prv}} = \{(x_i, y_i) \mid i \in [N_{\text{prv}}]\}$, loss function $\mathcal{L}(\theta, x_i)$, Parameters: learning rate $\eta_t$, noise scale $\sigma$, group size $L$, gradient norm bound $C$.
1: **Initialize** $\theta_0$ randomly
2: **for** $t \in [T]$ **do**
3:     Sample mini-batch $L_t$ with sampling probability $L/N$     ▷ Poisson sampling
4:     For each $i \in L_t$, compute $\mathbf{g}_t(x_i) \leftarrow \nabla_{\theta_t} \mathcal{L}(\theta_t, x_i)$     ▷ Compute gradient
5:     $\bar{\mathbf{g}}_t(x_i) \leftarrow \mathbf{g}_t(x_i) / \max\left(1, \frac{\|\mathbf{g}_t(x_i)\|_2}{C}\right)$     ▷ Clip gradient
6:     $\tilde{\mathbf{g}}_t \leftarrow \frac{1}{|L_t|}\left(\sum_i \bar{\mathbf{g}}_t(x_i) + \mathcal{N}\left(0, \sigma^2 C^2 \mathbf{I}\right)\right)$     ▷ Add noise
7:     $\theta_{t+1} \leftarrow \theta_t - \eta_t \tilde{\mathbf{g}}_t$     ▷ Descent
8: **Output** $\theta_T$ and compute the overall privacy cost $(\varepsilon, \delta)$ using a privacy accounting method.

---

## H  PARETO FRONTIERS FOR CLASSIFICATION WITH REJECT OPTION

In this section, we provide additional Pareto frontiers for classification with the reject option. In Figure 14, we show the Pareto frontier for the same task (gender classification) but for two different datasets. The Pareto frontiers are very similar with the exception of a difference in the baseline of accuracy for both models. In Figure 15a and Figure 15b we show the Pareto frontier for CelebA and CheXpert, respectively.

## I  EXTENDED RELATED WORK & GUIDANCE ON CHOICE OF FRAMEWORK

In the literature, different fairness notions have been implemented within DP-SGD and PATE frameworks.

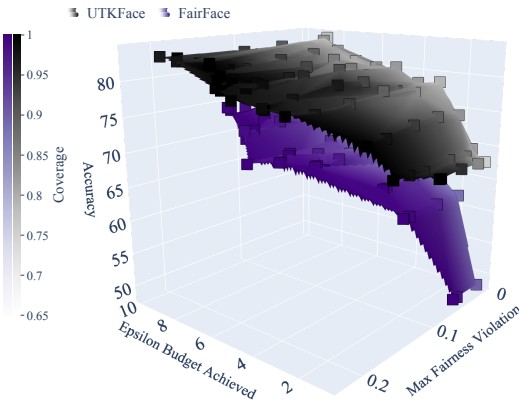

Figure 14: **Pareto Frontier of FairPATE results on UTKFace and FairFace.** The two surface have very similar shapes despite the differences in accuracy.

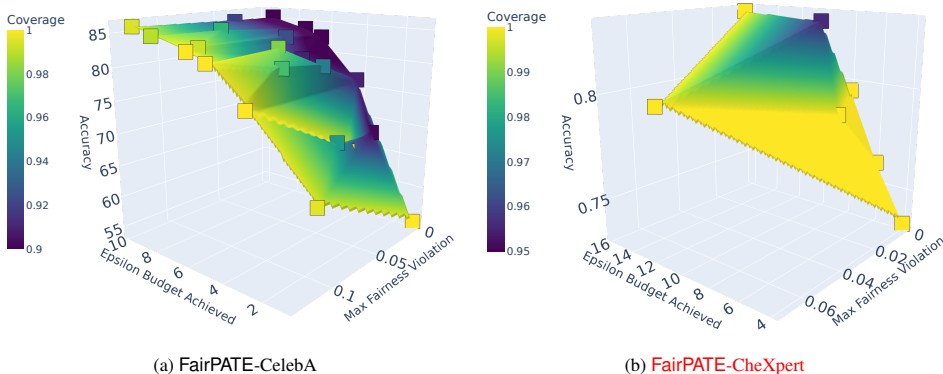

(a) FairPATE-CelebA

(b) FairPATE-CheXpert

Figure 15: **FairPATE on CelebA and CheXpert.** The figure plots the model results that are on the Pareto frontier. We observe that in both figures, accuracy increases with higher privacy budget $\varepsilon$, and that looser fairness constraints yield higher coverage.

The main contribution of FairPATE is to enable practical and $(\varepsilon, \delta)$-DP training with respect to all features and not just the sensitive ones in a PATE-type framework. Prior attempts at PATE-based frameworks such as Tran et al. (2022a;b) only consider the differential privacy w.r.t. to the sensitive attribute (see our discussion in Section 7 and Lowy et al. (2023, Definition 2.2) for an independent definition.

**Comparisons to other PATE-based models.** Tran et al. (2022a) study fairness properties of PATE and identified both algorithmic properties of the training (number of teachers, regularizer, privacy noise), and properties of the student data (magnitude of the input norm, and distance to the decision boundary) as factors influencing prediction fairness. To mitigate tensions, they proposed releasing the teacher models' prediction histogram as *soft labels* to train the student model. However, it has been shown that releasing the histograms leaks significant amounts of private information Wang et al. (2022), which makes their method leaks privacy above the promised DP guarantees. In contrast, in this work, we integrate fairness in the aggregation process while keeping the teachers' votes private, and, thereby providing the promised privacy guarantees.

**Comparisons to Tran et al. (2021b).** The authors proposed applying a Lagrangian dual approach for solving the joint optimization of fairness and privacy in ML. Therefore, they rely on a fairness constraint plus adaptive clipping and make the computations of the primal and dual update steps differentially private w.r.t. the considered sensitive attributes. However, their method adds a significant computational overhead, especially for larger ML models and mini-batch sizes (increase of up to factor 100).

**Comparisons to Kulynych et al. (2022).** The only prior work that we are aware of that has a sampling step for fairness within a private-learning setup is Kulynych et al. (2022). Our work implements a PATE mechanism for private learning whereas Kulynych et al. (2022) is DP-SGD based. This is crucial because the privacy analysis of the DP-SGD models is based on releasing every intermediate model weights and accumulating the privacy budget over samples (or mini-batches). Similar to us, Kulynych et al. (2022) adjust the probability of subgroups being sampled. But different from us, this probability goes to inform the Poisson distribution in the RDP accountant which then accumulates the privacy budget per mini-batch. As before, in this DP-SGD-based method, the privacy accounting happens at the weights-level whereas PATE's privacy analysis is done at the level of samples and labels.

Table 5 summarizes some of the frameworks we have highlighted in this section.

## I.1 CHOICE OF FRAMEWORK

The appropriate choice of privacy notion is application-dependent. However, a $(\epsilon, \delta)$-DP is a much stronger guarantee and it is by far the most prevalent notion of privacy. If interpretability is of concern, sample-space interventions are more appropriate since reasoning about the membership (or probability of membership) of a certain sample is easier than reasoning about high-dimensional gradients in the weight-space. Finally, all PATE-based frameworks assume access to an unlabeled public dataset due to its semi-supervised nature while DP-SGD-type methods do not require such dataset. This need can be addressed by using a subset of the training data, throwing out its labels, and employing it in place of the unlabeled dataset in PATE-based models. This is how the original PATE papers were evaluated, and we have evaluated FairPATE similarly.

In practice, however, the unlabeled dataset can come from a public dataset which is different than the sensitive training set. From a privacy accounting perspective only, the use of such public data is similar to the use of public data to pre-train DP-SGD models: in both cases, privacy cost, by definition, is not being accumulated for the public data. The difference is that pre-training requires labels, while the PATE's public dataset need not be labeled.

| Method | $(\epsilon, \delta)$-DP | DP w.r.t. Sensitive Attr. | DP Accounting for Fairness | Public Data | Fairness Intervention Space |
|---|---|---|---|---|---|
| FairPATE | ✓ | - | ✓ | Unlabeled | Samples |
| DP-FERMI | ✓ | - | ✓ | - | Weights |
| SF-PATE | - | ✓ | ✓ | Unlabeled | Weights |
| Jagielski et al. (2019) | - | ✓ | ✓ | - | Weights |
| Mozannar et al. (2020) | - | ✓ | ✓ | - | Weights |
| DP-IS SGD | ✓ | - | ✓ | - | Samples |
| Zhang et al. (2021) | ✓ | - | x | - | Optimization Hyper-parameters |

Table 5: **Summary of Various Fair and DP Training Models**, their privacy notion, their dependency on public data, whether they account for the privacy leakage of their fairness intervention, and the space where that intervention takes place

