# OpenReview forum: "FairPATE: Exposing the Pareto Frontier of Fairness, Privacy, Accuracy, and Coverage"
_ICLR.cc/2024/Conference — Submitted to ICLR 2024_

### Official Review · Reviewer_p5Rv · 2023-11-05

**Soundness:** 2 fair
**Presentation:** 3 good
**Contribution:** 2 fair
**Rating:** 3
**Confidence:** 4

**Summary:**

The paper discusses the problem of interactions between fairness, privacy, and accuracy constraints for PATE (Private Aggregation of Teacher Ensembles) type algorithms for differentially private learning. PATE style of algorithms first uses the private data (partitioned into several small partitions) to train base classifiers (teachers). Then, the algorithm uses these teachers to label some public data in a privacy preserving way. In particular, it cleverly chooses which points it can label without sacrificing too much privacy. Then another classifier (student) is trained on this newly privately labelled dataset, which is then released to the user. In this paper, this labelling step is used to also incorporate privacy constraints. Finally, the paper uses empirical evidence to suggest that their algorithm achieves a better privacy fairness accuracy trade-off than Loewy et. al. 2023.

**Strengths:**

* The paper is written quite clearly and is easily readable. The arguments of the authors come out clearly without ambiguity and the reader can easily follow the train-of-thought. I appreciated that very much.
* I also found the main algorithmic idea of this paper quite nice. The idea of that the algorithm chooses which points to label not only on the bassi of the privacy constraint but also the fairness constraint is quite neat and could be useful in other contexts. I appreciated this.

**Weaknesses:**

Despite the interesting idea of the paper, I am unable to support this paper for acceptance. The four main reasons are as follows (in decreasing order of severity).

1. __W1__ Significance: The paper aims to convey the significance of its contribution through experimental results (and I don't think there is anything wrong with it and in general support extensive empirical results), but the experiments present a rather bleak picture of the advantages of FairPATE.
    * **Minimal improvements** Most importantly, there are rarely any results where the improvements of PATE over baselines is larger than 1\%. For example in 1. Credit card dataset and 2. parkinson's dataset, there results differ by less than $1\%$.
    * **Several examples of underperformance** Several examples also show that FairPATE performs significantly worse than competitor. Examples include Demographic Parity for Adult dataset and UTK Face ($\epsilon=5$).
    * **Misleading regarding diversity of experiments** In the introduction, the paper sells itself regarding performing a wide range of experiments but results on nearly half of these datasets are hidden away in the appendix without any mention in the main text and without comparisons. Experiments on **CheXpert, CelebA, FairFace, and Retired-Adults** are not available in the main text and it is very hard to interpret the results presented for this in the appendix.  In fact, there are **no results of FAIR-PATE on CheXpert** in the paper
    * Please show comparison on these for demographic parity and equalized odds (similar to Figure 2,3 with comparison to Loewe et. al.'s algorithm) on these four datasets. In addition, there are results in the literature that run DP algorithms on CelebA and CheXpert. Comparisons should be made to them to show what is the sacrifice being made compared to state-of-the-art.

2. __ W2__ **Wrong Conclusion from Theorem 2** I did not go through the detail of Theorem 1 but I assume it is correct and follows from Group privacy arguments. However, the sentence above Theorem 1 (Point C2) claims that the result proves that  " pre-processing ... will necesarrily degrade the guarantee of the .... private learning mechanism". How ever this is not what the Theorem shows. The theorem only proves a privacy guarantee of $M\odot P_{\text{pre}}$ not that this is the tightest privacy guarantee possible for the composed mechanism. Am I missing something ?

3. __W3__ **Abstaining from prediction for fairness reasons** The introduction justifies this as _"if a decision cannot be made without violating a pre-specified fairness metric, then the model can refuse to answer at which point the decision can be relegated to a human judge"_. If I have understood this correctly, this is a flawed argument. For example, consider the situation where there are group of data points from the majority group, on which if the algorithm predicts correctly the fairness will be violated and hence the algorithm abstains. Nevertheless, the prediction is easy for this group and the judge nevertheless predicts on them correctly and the overall fairness is still violated. Isn't this pointless then to relegate this to the judge ? Intuitively, it appears that any problem can be made fair this way by simply refusing to predict on certain majority groups thereby artificially boosting the fairness of the algorithm. The correct fairness solution should aim to improve performance on the minority group instead.


4. __W4__  **Unfair Comparisons** Fair-PATE  and Loewy et. al's algorithm do not work under the same restrictions of differential privacy. Specifically, Fair-PATE uses public unlabelled data and Loewy et. al doesn't and we know that (even a small amount of) public data can severely help in improving accuracy of DP models (perhaps fairness too). Hence, it appears to me that Loewy et. al. uses a significantly stricter notion of privacy and achieves nearly comparable result and in some cases even better (for tabular datasets, there are no comparison on vision datasets with Loewy et. al.).

**Questions:**

* **Motivation** I understand that there are several works showing that privacy incurs a sacrifice in fairness. However, one possible approach to this problem is simply improving the accuracy and perhaps the resultant accuracy also leads to high fairness. In fact Berrada et. al. (2023) makes this argument with some experiments in their paper. I did not see an argument in this paper why this approach is not expected to solve all problems. Why should there be a trade-off between privacy and fairness ? Is it known in the literature and are there theoretical studies arguing about the necessity of a trade-off ?

* **Missing baselines** Why is there no comparison with the algorithm of Zhang et. al. (2021), Kulynuych et. al. (2022) and Berrada et. al. (2023) ? They are mentioned in one or two sentences at the very end of the paper but without any comprehensive empirical comparison.
* **FairDP-SGD" What is FairDP-SGD ? It doesn't seem to have been defined anywhere ? Is it an existing work ? Where is FAIR-PATE's result on CheXpert ?

* In addition to this, please also also address __W1,W2,W3,W4__.

---

> ### Author Response · Authors · 2023-11-18
> **Response to Weaknesses (Part 1)**
>
> We thank the reviewer for their feedback. Below we address each point raised in "Weaknesses" section.
>
> - **Several examples of underperformance** Several examples also show that FairPATE performs significantly worse than competitor. Examples include Demographic Parity for Adult dataset and UTK Face ().
>
> 	- For the concerns about significance, please see our General Response.
>
> 	- We have noted clearly in the main paper that for the Adult dataset, under Demographic Parity DP-Fermi performs better than FairPATE for smaller epsilon values (Figure 10). However that trend is not consistent over different datasets (on UTKFace, FairPATE does better for lower epsilons and can even go lower in terms of achievable classification error—see Figure 6). We provide a discussion on the cause of these performance differentials in response to **Reviewer XkhF.**
>
> - **Misleading regarding diversity of experiments** In the introduction, the paper sells itself regarding performing a wide range of experiments but results on nearly half of these datasets are hidden away in the appendix without any mention in the main text and without comparisons. Experiments on CheXpert, CelebA, FairFace, and Retired-Adults are not available in the main text and it is very hard to interpret the results presented for this in the appendix. In fact, there areno results of FAIR-PATE on CheXpert in the paper
>
> 	- The CheXpert results were already present in the paper (in the last figure of the appendix) but was unfortunately mislabeled. We apologize for the error.  The figure's label has been corrected in the updated manuscript.
>
> 	- We currently have results of FairPATE on CheXpert, CelebA and FairFace in Section H in the appendix.  Unfortunately given the space constraints we cannot add all the aforementioned results in the main text.
>
> - Please show comparison on these for demographic parity and equalized odds (similar to Figure 2,3 with comparison to Loewe et. al.'s algorithm) on these four datasets. In addition, there are results in the literature that run DP algorithms on CelebA and CheXpert. Comparisons should be made to them to show what is the sacrifice being made compared to state-of-the-art.
>
> 	- DP-FERMI is an expensive algorithm to run. On CelebA, it requires 33 hours to train one model for 200 epochs as specified in the paper without parameter tuning on an NVIDIA A100; while the model has at least 4 hyper-parameters to tune. Nevertheless, we are currently in the process on running DP-FERMI on CelebA, and we will report the results as we acquire them in the coming days.
>
> - **W2 Wrong Conclusion from Theorem 2** I did not go through the detail of Theorem 1 but I assume it is correct and follows from Group privacy arguments. However, the sentence above Theorem 1 (Point C2) claims that the result proves that " pre-processing ... will necesarrily degrade the guarantee of the .... private learning mechanism". How ever this is not what the Theorem shows. The theorem only proves a privacy guarantee of   $M \odot P_{\mathrm{pre}}$ not that this is the tightest privacy guarantee possible for the composed mechanism. Am I missing something ?
>
> 	- We are not sure we follow the reviewer's comments here. Does the reviewer take issue with the phrasing of the claim " pre-processing ... will necesarrily degrade the guarantee of the .... private learning mechanism"?
>
> 	- If yes, then we apologize for the confusion. We can adjust the claim as
> 	  > pre-processing ... will necessarily degrade the guarantee of the .... composed preprocessor and private learning mechanism
> 	- Of course, the private learning algorithm does not degrade on its own; but via the inclusion of a fairness pre-processor, we have shown that the joint fair and private mechanism will have degraded privacy guarantee which is always an upper bound on the privacy leakage
>
> 	- We are also unsure of what the reviewer mean by the "tightest privacy guarantee possible for the composed mechanism." Do they mean that another $\mathcal{P}_\text{pre}$ could possibly reduce the privacy cost? If yes, then given that the pre-processor is simply ensuring the fairness definition holds (via a filter that class balances the dataset conditioned on the sensitive attribute) can the reviewer recommend another pre-processor?

---

> > ### Comment · Reviewer_p5Rv · 2023-11-22
> >
> > I thank the authors for their response. I have read through the entire comment in response to my questions. I will try to keep my replies brief and provide suggestions on what I think can improve the manuscript in subsequent drafts.
> >
> > [W1]  Currently the paper presents a new algorithm for improving fairness-accuracy pareto frontier but the main weakness  here is that in a significant number of situations this improvement does not happen but rather a degradation happens. I would recommend that the authors rethink about what the main contribution is, perhaps identify more clearly in the main text where they expect an improvement and where they don't and then show that through experiments. Doing this discussion through rebuttal is not the best as this a major point of this paper.
> >
> > Moreover, i had noticed the plots in Appendix H, but the plots were not very informative regarding how important the improvement is due to the missing comparisons. Again I appreciate that the authors are running the experiments but this is an important contribution of the paper and hence the experiments for this should not really be done during the rebuttal and it should be possible to present the "improvements of the algorithm on seven distinct datasets" in the main paper. Perhaps, the paper needs a restructuring if all the exerimental improvements obtained by a novel algorithm cannot be presented in the main pages of the paper.

---

> > > ### Comment · Reviewer_p5Rv · 2023-11-22
> > > **Response to Weakness 2**
> > >
> > > My point in weakness 2 is that the result shows that the composed mechanism is at least $(\epsilon,\delta)$ DP.
> > >
> > > However the argument in the text says that the "will necessarily degrade the guarantee".
> > >
> > > The first sentence is an upper bound on the privacy guarantee of the mechanism whereas the second sentence is a sentence akin to a lower bound.

---

> > > ### Author Response · Authors · 2023-11-23
> > >
> > > We thank the reviewer for engaging with us. We appreciate the time and effort you have dedicated to reviewing our work.
> > >
> > >
> > > - [W1] Currently the paper presents a new algorithm for improving fairness-accuracy pareto frontier but the main weakness here is that in a significant number of situations this improvement does not happen but rather a degradation happens. I would recommend that the authors rethink about what the main contribution is, perhaps identify more clearly in the main text where they expect an improvement and where they don't and then show that through experiments. Doing this discussion through rebuttal is not the best as this a major point of this paper.
> > >
> > > - Moreover, i had noticed the plots in Appendix H, but the plots were not very informative regarding how important the improvement is due to the missing comparisons. Again I appreciate that the authors are running the experiments but this is an important contribution of the paper and hence the experiments for this should not really be done during the rebuttal and it should be possible to present the "improvements of the algorithm on seven distinct datasets" in the main paper. Perhaps, the paper needs a restructuring if all the exerimental improvements obtained by a novel algorithm cannot be presented in the main pages of the paper.
> > >
> > > 	- We thank the reviewer for their feedback. We made a general statement regarding our contributions (in the Author General Response). Please also consider our edits and response to **reviewer Xkhf** newest comments.
> > >
> > > - My point in weakness 2 is that the result shows that the composed mechanism is at least  $(\epsilon, \delta)$ -DP. However the argument in the text says that the "will necessarily degrade the guarantee". The first sentence is an upper bound on the privacy guarantee of the mechanism whereas the second sentence is a sentence akin to a lower bound.
> > >
> > > 	- We have edited the sentence in question for clarity to:
> > > 	  > (C2) Fairness pre-processing increases the cost of private training. In Theorem 1, we show that pre-processing the training data to equalize subpopulation rates will degrade the privacy guarantee of the composed mechanism (i.e., preprocessor followed by private learning) compared to that of the private learning mechanism alone
> > > - I am a bit confused with what the authors mean by "Please note that IPP is an inference-time algorithm. It is incorrect to say that the *"algorithm predicts correctly that the fairness will be violated because there is no prediction involved in this process"* and *"Since IPP is releasing decisions, it can keep an exact record of the decisions released (this is what counter keeps in Algorithm 1). Therefore, the rates calculated based on this record is exact and so are the detected violations that would prompt the algorithm to reject a certain query."*
> > >   If the model is creating an output, which is then judged by IPP whether to release or not - isn't this an example of prediction ? Or pehaps I am misunderstanding something
> > > 	- When IPP is employed (and it can be employed for any model, not just FairPATE) then, a binary classification task $X \mapsto \{0, 1\}$ is turned into a selective classification $X \mapsto \{0, 1, \bot\}$ where $\bot$ means that no prediction is released (a query is not answered). Therefore, the answer to the question is no. If the query is rejected, it is as if no prediction has taken place. For instance, this query will not be considered when measuring accuracy—but it will be considered when measuring coverage.
> > >
> > > - Regarding W4, can the authors clarify whether the number of **private** data (excluding public unlabelled data) points seen by FairPATE and Loewy et. al. are exactly same ?
> > >
> > > 	- We believe we have answered this question in the follow-up in the general comment. The answer is no because evaluating any PATE model involves throwing out labels for a subset of the training set that is going to function as the unlabeled set. Note two things: first, **this is not in the favor of the PATE mechanism (because we are losing label information)**, two, this is not a choice given the semi-supervised nature of PATE.
> > >
> > > 	  For sake of completeness, please consider the alternative scenario where we have ensured the premise of the question (that is we have exactly equal number of private samples for both DP-SGD and PATE), then this means that we should find another dataset to use as unlabeled dataset in PATE. Under this evaluation scheme, the PATE model receives strictly more data samples. **We find this evaluation scheme unfair to supervised models such as DP-SGD.**

---

> > > > ### Comment · Reviewer_p5Rv · 2023-11-23
> > > >
> > > > I thank the authors for engaging.
> > > >
> > > > Regarding W1 and W2. I maintain my concern about W1 that this is not something that can solved with minor edits and clarification but needs restructuring. The comments here in the rebuttal is a good place to start in my opinion.
> > > >
> > > > Regarding W2, if the authors agree with the difference between what the result conveys and what a lower bound would convey, perhaps the author can include a lower bound result in the work to make their original point, which is I think important to the work that pre-processing indeed necessarily costs privacy.
> > > >
> > > > Regarding IPP, I do not agree with the characterisation. This is prediction with abstention or selective classification, as the authors describe. And it is not clear to me from the text, how IPP guarantees that it will not simply relegate examples that are 1) easy to classify 2) and classification will lead to unfairness to the human classifier even though there are other examples in the test set which are more nuanced and should have been relegated to the human classifier. An easy-to-see instance is that first the model can do usual classification without abstention, then check its fairness, then figure out which examples it can refuse to classify in order to stay within this budget (note that there can be many such sets; the algorithm can choose a set randomly), and then abstain from predicting for these.
> > > >
> > > > Finally, regarding W4, I see the stress the authors are putting on the absence of labels and using it to say that PATE sees stricly less information than Loewy et. al. I also reject this characterisation by arguing that the PATE algorithm seems (unlabelled) data points whose privacy it does not need to preserve whereas Loewy et. al. needs to preserve privacy of every data point it sees.

---

> ### Author Response · Authors · 2023-11-18
> **Response to Weaknesses (Part 2)**
>
> - **W3** **Abstaining from prediction for fairness reasons** The introduction justifies this as *"if a decision cannot be made without violating a pre-specified fairness metric, then the model can refuse to answer at which point the decision can be relegated to a human judge"*. If I have understood this correctly, this is a flawed argument...
>
> 	- We think there has been a misunderstanding regarding a) purpose of the reject option for fairness, and b) the inference-time nature of the IPP algorithm that implements it. What the reject-option enables is for a human (e.g. a trained judge) to take over the decision making in case of a fairness constraint violation. **The presumption here is that a trained judge is better placed to make impactful decisions than the algorithm.** Therefore, the human judge can choose to accept violations of the fairness constraint on a case-by-case basis. We do not believe the human judge should be placed under the same constraints as the algorithm given the vast difference in training, experience and accountability faced by the human judge versus the algorithm.
>
> -  Consider the situation where there are group of data points from the majority group, on which if the algorithm predicts correctly the fairness will be violated and hence the algorithm abstains
> 	- Please note that IPP is an **inference-time** algorithm. It is incorrect to say that the "algorithm predicts correctly that the fairness will be violated" because there is no prediction involved in this process. For instance, using Demographic Parity, the fairness metric requires that the rates of acceptance be similar for different subgroups. Since IPP is releasing decisions, it can keep an exact record of the decisions released (this is what $m(z,k)$ counter keeps in Algorithm 1). Therefore, the rates calculated based on this record is exact and so are the detected violations that would prompt the algorithm to reject a certain query.
>
> - Intuitively, it appears that any problem can be made fair this way by simply refusing to predict on certain majority groups thereby artificially boosting the fairness of the algorithm. The correct fairness solution should aim to improve performance on the minority group instead.
> 	- Our standard fairness metric does not require a notion of majority vs. minority. At any given decision point, our algorithm (IPP) bounds the worst-case fairness violations experienced by any subpopulation. Furthermore, the reject-option is not a silver bullet; as it comes with a trade-off with coverage (1-rejection rate). Therefore the more rejections translate more work for the human judge. It is, therefore, in the best interest of deployer of the upstream algorithm (FairPATE or DP-FERMI, or any other fairness-aware) algorithm to minimize fairness violations at training-time; because an unfair algorithm will increase the work that human judges would need to do— eliminating the need for the fair algorithm in the first place.
>
> - **W4**  **Unfair Comparisons** Fair-PATE and Loewy et. al's algorithm do not work under the same restrictions of differential privacy. Specifically, Fair-PATE uses public unlabelled data and Loewy et. al doesn't and we know that (even a small amount of) public data can severely help in improving accuracy of DP models (perhaps fairness too). Hence, it appears to me that Loewy et. al. uses a significantly stricter notion of privacy and achieves nearly comparable result and in some cases even better (for tabular datasets, there are no comparison on vision datasets with Loewy et. al.).
>
> 	- We wish to clarify and disentangle two issues that the reviewer points out: a) access to public data, b) access to additional data. All of our baselines experiments are conducted using the exact data splits of Lowy et al. 2023. This means that we do not use any additional training data compared to our baselines. Instead, we use the training data available to us and partition it into public and private. Consistent with the original PATE framework, we do not use the labels of the public dataset; instead we train teacher models on the private partition and label the public points under privacy and fairness constraints. In Figure 1.a, a DP-SGD method like Lowy et al. 2023 replaces the shaded Private Model. Therefore, we respectfully disagree with the reviewer's assessment that the comparisons are not fair data-wise. As a matter of fact, in our experiments, DP-Fermi receives additional information compared to FairPATE (namely, the labels of the public data partition).

---

> > ### Comment · Reviewer_p5Rv · 2023-11-22
> >
> > I am a bit confused with what the authors mean by "Please note that IPP is an inference-time algorithm. It is incorrect to say that the _"algorithm predicts correctly that the fairness will be violated because there is no prediction involved in this process"_ and _"Since IPP is releasing decisions, it can keep an exact record of the decisions released (this is what
> >  counter keeps in Algorithm 1). Therefore, the rates calculated based on this record is exact and so are the detected violations that would prompt the algorithm to reject a certain query."_
> >
> > If the model is creating an output, which is then judged by IPP whether to release or not - isn't this an example of prediction ? Or pehaps I am misunderstanding something

---

> ### Author Response · Authors · 2023-11-18
> **Response To Questions**
>
> - Motivation. I understand that there are several works showing that privacy incurs a sacrifice in fairness. However, one possible approach to this problem is simply improving the accuracy and perhaps the resultant accuracy also leads to high fairness. In fact Berrada et. al. (2023) makes this argument with some experiments in their paper. I did not see an argument in this paper why this approach is not expected to solve all problems. Why should there be a trade-off between privacy and fairness ? Is it known in the literature and are there theoretical studies arguing about the necessity of a trade-off ?
>
> 	- Berrada et al. 2023 only consider unfairness with respect to loss disparity. This particular fairness metric is inherently a measure of the generalization of the model. That is, if a model is well generalized, then it is also well-generalized for different subpopulations of the data; therefore, it is only natural that it would exhibit lower loss disparity. As a result Berrada et al. 2023 do not consider (or rather, need to consider) an unfairness mitigation step. Our results (and that of Lowy et al. 2023 and others) show that for other group fairness metrics, this is not the case. That is, fairness and accuracy do not align so well all the time.
>
> 	- In terms of theoretical results Cummings et al. 2019 provide an the impossibility result for DP learning and exact fairness; while also demonstrating a positive result for the case of approximate fairness (our settings). More recently, Mangold et al. 2023 have showed that "the fairness (and accuracy) costs induced by privacy in differentially private classification vanishes at a $O(\sqrt{p}/n)$ rate, where n is the number of training records, and p the number of parameters." This is achieved through a proof of pointwise Lipschitz smoothness of group fairness metrics with respect to the model where the pointwise Lipschitz constant explicitly depends on the confidence margin of the model, and may be different for each sensitive group. Therefore, in the presence of minority groups in the data, we can expect a non-zero gap between private and non-private models.
>
> - Missing baselines. Why is there no comparison with the algorithm of Zhang et. al. (2021), Kulynuych et. al. (2022) and Berrada et. al. (2023) ? They are mentioned in one or two sentences at the very end of the paper but without any comprehensive empirical comparison.
>
> 	- Kulynych et. al 2021 does not account for the privacy leakage of its importance sampling step therefore a fair comparison with methods such as ours or Lowy et al. 2023 is not possible
>
> 	- Berrada et al. 2023 only considers loss parity, and has no unfairness mitigation for two of our gorup fairness metrics (see our previous answer for more details)
>
> 	- Zhang et al. 2021 implement early stopping to save on privacy budget; however, they do not measure the privacy leakage of their early stopping criterion therefore the privacy loss reported is underestimated
>
> - What is FairDP-SGD ? It doesn't seem to have been defined anywhere ? Is it an existing work ? Where is FAIR-PATE's result on CheXpert ?
>
> 	- This is an error on our part. The figure in the appendix was mislabelled. We apologize for the confusion. The figure's label has been corrected in the updated manuscript.
>
> We once again thank the reviewer for their detailed feedback and kindly ask them to consider raising their score if we have addressed their concerns successfully. We are happy to answer further questions.

---

> > ### Comment · Reviewer_p5Rv · 2023-11-22
> >
> > Regarding W4, can the authors clarify whether the number of __private__ data (excluding public unlabelled data) points seen by FairPATE and Loewy et. al. are exactly same ?

---

> > ### Comment · Reviewer_p5Rv · 2023-11-22
> > **Regarading existing literature**
> >
> > * Kulynych et. al. - Do the authors mean that Algorithm in Kulynych et. al. (which does DP importance sampling) does not respect the $(\epsilon,\delta)$-DP ?
> > * Berrada et. al. - Perhaps the authors can then compare their method with the algorithm in berrada et. al. and show that Berrada et. al.'s method indeed suffers in DP and EO. This would be a strong and important point to make showing the necessity of this algorithm.
> > * I am a bit confused what the authors mean about Cummings et. al's result. If their paper shows a negative result about pure DP but positive result about Approximate DP, then this is arguing against the existence of unfairness problem in approx DP ?

---

> > > ### Author Response · Authors · 2023-11-23
> > >
> > > - Kulynych et. al. - Do the authors mean that Algorithm in Kulynych et. al. (which does DP importance sampling) does not respect the  $(\epsilon, \delta)$ -DP ?
> > >
> > > 	- No, their mechanism remains $(\epsilon, \delta)$ -DP private (per Lemma B.7). We also wish to correct a prior assertion that they do not account for the cost of their importance sampling. They do. We reached out to one of the authors and quote them here verbatim:
> > > 	  > In the sense of “privacy cost” as in your Theorem 1, in DP-IS-SGD you are indeed also paying additional cost for importance sampling. In DP-IS-SGD, we set the mini-batch sampling probability for privacy accounting to $p^* \propto \frac{1}{n_G}$ , which grows as $n_G$ gets smaller, where $n_G$ is the size of the smallest group. (The privacy guarantee deteriorates when the mini-batch sampling probability increases given all other parameters are fixed.) As a result, you would likely need to add more noise in DP-IS-SGD for the same $(\epsilon, \delta)$ . DP-IS-SGD can be effective when groups are not too small or when the effects of reducing disparate impact through importance sampling outweigh the additional privacy costs.
> > > 	- Berrada et. al. - Perhaps the authors can then compare their method with the algorithm in Berrada et. al. and show that Berrada et. al.'s method indeed suffers in DP and EO. This would be a strong and important point to make showing the necessity of this algorithm.
> > >
> > > 		- We note that Berrada does not introduce a novel method. Their main contention is that DP-SGD models trained to good generalization levels  (e.g. trained with large enough batch sizes, etc.)  do not exhibit large loss disparity.  We previously argued that this group fairness notion (Loss Parity) is essentially a generalization notion which is why Berrada et al's claims are reasonable. Unfortunately, they have not released any code or data, and we cannot be expected to replicate their industrial-scale experiments due to computation and resource constraints.
> > >
> > > 		- Having said that, we have run more experiments with the error parity notion using our test bench and indeed we observe that compared to other group fairness notions (demographic parity and equality of odds), error parity results are the least affected by the introduction of differential privacy training. **We report the new results in Section E.3 of the revised manuscript.**
> > >
> > > 	- I am a bit confused what the authors mean about Cummings et. al's result. If their paper shows a negative result about pure DP but positive result about Approximate DP, then this is arguing against the existence of unfairness problem in approx DP ?
> > >
> > > 		- No. What Cummings defines as approximate fairness is non-zero fairness violations ( $\gamma > 0$ ). In other words, Cummings' is a theoretical result that does not differentiate between different positive $\gamma$ values, as long as they are non-zero. In Theorem 2, they show that the bound on the number of samples needed to learn a hypothesis  (in a PAC sense) in a differentially private way, has an inverse relationship with $\gamma$ (they use the notation $\alpha$ instead); meaning that the smaller the $\gamma$ the more samples it would be needed to satisfy this existence condition. In the context of your question, the negative result is more meaningful which proves eliminating the unfairness problem altogether is impossible.
> > >
> > > Thank you

---

> > > > ### Comment · Reviewer_p5Rv · 2023-11-23
> > > >
> > > > Thank you for the correction. I would recommend adding an "addendum" to the previous comment so that people reading this conversation do not get the wrong information.
> > > >
> > > > I would also recommend adding your own comparisons and analysis in the paper about Kulynych et. al. as opposed to quoting the authors, in particular empirical comparisons.
> > > >
> > > > Thank you for adding E.3, I beleive this should go in the paper and you can compare their "extended DP-SGD" with your algorithm on all the vision datasets you use for all three metrics.
> > > >
> > > > Finally, Regarding Cummings et. al. result, I agree their negative result is more important which is why i was surprised to see your mention their positive result in the previous message. However, their negative result only deals with pure DP and not with approximate DP and I would recommend searching for a negative result from literature that also shows a trade-off with privacy and fairness.

---

### Official Review · Reviewer_Mex5 · 2023-11-06

**Soundness:** 3 good
**Presentation:** 3 good
**Contribution:** 1 poor
**Rating:** 3
**Confidence:** 3

**Summary:**

The work considers the inclusion of fairness constraints into a method for differentially private
(DP) training (or data generation from private data) based on transfer learning. The paper argues
that in this "PATE" approach which accounts of privacy concerns using DP, there is only one
sensible place to incorporate fairness using an intervention (i.e. adjusting what/whether) data
proceeds to subsequent PATE steps. This step is the point after the transfer from an ensemble of
teachers is made. The paper puts a mechanism there that will reject some queries/instances if they
result in violations of fairness which is a function of all of the prior decisions of the
mechanism. The work evaluates this approach relative to 2 other DP-based systems that incorporate
fairness showing mostly preferable trade-offs between fairness, accuracy, and privacy; though this
benefit is small.

**Strengths:**

+ Fairly well written and easy to follow.

+ The points of intervention discussions give a nice overview of the PATE approach and ways in
  which additional mechanisms can be independently injected. Note, however, the independent
  intervention assumption is a weakness below.

+ Rejection for fairness does give additional options for achieving fairness though this too comes
  with a weakness below.

**Weaknesses:**

- The implications of rejecting for fairness are not considered. Rejection for privacy has
  implications in terms of privacy budget and likewise rejections for fairness come with
  implications and ignoring them might be responsible for the observed gains on the Pareto
  frontier. Consider the noted rejection example:

    "If at inference-time a decision cannot be made without violating a pre-specified fairness
     metric, then the model can refuse to answer, at which point that decision could be relegated
     to a human judge"

  The important implication here is that there will still be a judgement; it is just that the model
  will not be making it. Regardless of whether the result of the human judgement will produce fair
  or unfair overall statistics (that consider ultimate judgement whether by model or human), those
  decisions need to be incorporated into subsequent fairness calculus. Even if a query is rejected
  due to privacy, and if a decision is made for it subsequently, it would need to be accounted for
  in subsequent fairness decisions.

  Suggestion: incorporate ultimate decisions, whether by model or human, into the rejection
  mechanism; i.e. update counts m(z, k) based on human decisions. Given that humans might put the
  group counts into already violating territory, it may be necessary to rewrite Line 7 of Algorithm
  1 to check whether the fairness criterion is improving or not due to the decision and allow
  queries that improve statistics even though those statistics already violate γ threshold.
  Handling rejection in experiments will also need to be done but unsure what the best approach
  there would be. Perhaps a random human decision maker?

- In arguments for intervention points, assumptions are made which preclude solutions. They assume
  the intervention need to be made independent of other mechanisms in PATE. That is, they cannot
  consider information internal to decision making that is not described by Figure 1 like
  individual teacher outputs. This leaves the possibility that some fairness methods might be able
  to integrated with PATE in a closer manner than the options described. One example is that they
  might include the teacher outputs instead of operating on the overall predicted class like
  Algorithm 1 assumes presently. C3 in particular suggests that some interventions will not account
  for privacy budget correctly due to special circumstances and suggests at Point 4, they can be
  budgeting can be handled correctly. Nothing is stopping a design from refunding privacy budget if
  a query is rejected subsequently to an intervention point.

  Suggestion: rephrase arguments for why some intervention points are bad to make sure they don't
  also make assumptions about how the interventions are made and whether they can interact with
  privacy budget.

- Results in the Pareto frontier show small improvements, no improvements, and in some cases worse
  results than prior baselines.

  Suggestion: Include more experimental samples in the results to make sure the statistical
  validity of any improvement claims is good. This may require larger datasets. Related, the
  experiments show error bars but how they are derived is not explained.

- Comparisons against methods in which rejection due to fairness is not an option may not be fair.

  Suggestion: either integrate suggestion regarding accounting for rejection above, or incorporate
  some form of rejection (or simulate it) in the existing methods being compared to. It may be that
  the best methodology is not FairPATE but some existing baselines if adjusted to include fairness
  rejection option.

Smaller things:

- Rejection rate is not shown in any experiments. One could view a misclassification as a
  rejection, however. Please include rejection rates or view them as misclassifications in the
  results.

- The distribution whose fairness need to be protected is left to be guessed by the reader. For
  privacy, it is more clear that it is the private data that is sensitive and thus privacy
  budgeting is done when accessing that private data as opposed to the public data. For fairness,
  the impact on individuals in the private dataset seems to be non-existent as the decisions for
  them are never made, released, or implemented in some downstream outcome. I presume, then, it is
  the fairness needs to be respected on the public data.

  Algorithm 1 and several points throughout the work hint at this. However, there is also the
  consideration of intervention points 1,2,3 which seem odd as they points seen before any
  individual for whom fairness is considered is seen. That is, fairness about public individuals
  cannot be made there, independent of any other issues such as privacy budgeting. Further, Theorem
  1 discusses a demographic parity pre-processor which achieves demographic parity on private data
  which I presume is irrelevant.

- The statement

    "PATE relies on unlabeled public data, which lacks the ground truth labels Y"

  is a bit confusing unless one has already understood that fairness is with respect to public
  data. PATE also relies on private labeled data to create the teachers.

- The Privacy Analysis paragraph could be greatly simplified to just the last sentence regarding
  post-processing.

Smallest things:

- Double "violations" near "violations of demographic disparity violations".

- The statement "DP that only protects privacy of a given sensitive feature" might be
  mischaracterizing DP. It is not focused on features or even data but rather the impact of
  *individuals* on visible results.

**Questions:**

Question A: Is reasonable to ignore downstream decisions from queries rejected due to fairness
  (i.e. contrary to my suggestion in the weaknesses above)?

Question B: C1 makes a point that adding privacy after fairness may break fairness. What about in
  expectation? Were one to view the demographic statistics defining fairness measures in
  expectations, wouldn't they remain fair?

Question C: Theorem 1 makes a statement about a pre-processor inducing privacy parameter
  degradation but FairPATE (or PATE) appears to fit the definition of a pre-processor. If the point
  of the Theorem is to argue against pre-processors, isn't it also arguing against PATE/FairPATE?
  Unrelated, what is "ordering defined over the input space X" and why is it necessary?

---

> ### Author Response · Authors · 2023-11-18
> **Response to Weaknesses**
>
> We thank the reviewer for their feedback. Below we address each point raised in "Weaknesses" section. We kindly note that in the following, we have grouped the reviewer's comments that addressed a common concern:
>
> - The implications of rejecting for fairness are not considered. Rejection for privacy has implications in terms of privacy budget and likewise rejections for fairness come with implications and ignoring them might be responsible for the observed gains on the Pareto frontier. Consider the noted rejection example: ...
>
> 	- Our understanding of the reviewer's remark here is that in a human-in-the-loop setting, the mistakes of the human would ultimately be mistakes of the end-to-end system; and therefore, measuring the efficacy of the system should consider the human error as well. While we agree with the reviewer, we believe that measuring the end-to-end error of the human-in-the-loop system should be the topic of a future study.
>
> 	- We have added the following to our discussion in Section 7:
> 	  > We note that in a human-in-the-loop system, the mistakes of the human would ultimately be mistakes of the end-to-end system; and therefore, measuring the efficacy of the system should consider the human error as well. We have shown that FairPATE and IPP enable such applications but whether such a system is the appropriate choice for a given application is out of the scope of the current study.
>
> - In arguments for intervention points, assumptions are made which preclude solutions. They assume the intervention need to be made independent of other mechanisms in PATE. That is, they cannot consider information internal to decision making that is not described by Figure 1 like individual teacher outputs. This leaves the possibility that some fairness methods might be able to integrated with PATE in a closer manner than the options described. One example is that they might include the teacher outputs instead of operating on the overall predicted class like Algorithm 1 assumes presently. C3 in particular suggests that some interventions will not account for privacy budget correctly due to special circumstances and suggests at Point 4, they can be budgeting can be handled correctly. Nothing is stopping a design from refunding privacy budget if a query is rejected subsequently to an intervention point.
>
> 	- Regarding the suggested alternative scheme, we would like to to ensure that we understand the premise correctly: is the reviewer suggesting that we run PATE, accept and reject queries according to standard PATE analysis, and then apply the fairness intervention and reject possibly some more queries; and come back to give back some of the privacy budget?
>
> 	- If the answer is yes, then we note that standard PATE privacy analysis in Papernot et al. 2016 (summarized  in Section B in the appendix) is based on the probability of answering a certain query. To recoup the "unused" privacy budget one needs to find the difference between the privacy budget if the sample would have been answered and the privacy budget if the sample wouldn't have been answered. Note that the this is exactly what FairPATE already performs. Therefore, the suggested scheme is functionally equivalent to FairPATE but presents additional complexity.
>
> - Results in the Pareto frontier show small improvements, no improvements, and in some cases worse results than prior baselines.
> - Suggestion: Include more experimental samples in the results to make sure the statistical validity of any improvement claims is good. This may require larger datasets. Related, the experiments show error bars but how they are derived is not explained.
>
> 	- We have included more results in the revised manuscript. Please see our responses to **Reviewers XkhF and p5Rv.**
>
> - Comparisons against methods in which rejection due to fairness is not an option may not be fair.
> - Suggestion: either integrate suggestion regarding accounting for rejection above, or incorporate some form of rejection (or simulate it) in the existing methods being compared to. It may be that the best methodology is not FairPATE but some existing baselines if adjusted to include fairness rejection option.
>
> 	- The reviewer is correct that the IPP algorithm is an inference-time algorithm, therefore, its use is not limited to a FairPATE model.
> 	- We have run additional experiments on other models with the IPP;  and included the results in Section E.1 in the appendix. We find that FairPATE with IPP outperforms baselines (with IPP) in most regions fairness-privacy settings in both accuracy and coverage. See Figure 11 for a 2d Pareto surface, or Figure 12 for a corresponding 3d plot which showcases the accuracy performance improvements better.

---

> ### Author Response · Authors · 2023-11-18
> **Response to Weaknesses (Part 2)**
>
> Smaller things:
> - Rejection rate is not shown in any experiments. One could view a misclassification as a rejection, however. Please include rejection rates or view them as misclassifications in the results.
>
> 	- We do report **coverage** which is 1-rejection rate for the results that use the IPP. See Figure 5 where coverage is marked by the **color**; similarly the Pareto frontiers in the appendix all include coverage encoded in color.
>
> - The distribution whose fairness need to be protected is left to be guessed by the reader. For privacy, it is more clear that it is the private data that is sensitive and thus privacy budgeting is done when accessing that private data as opposed to the public data. For fairness, the impact on individuals in the private dataset seems to be non-existent as the decisions for them are never made, released, or implemented in some downstream outcome. I presume, then, it is the fairness needs to be respected on the public data.
> - Algorithm 1 and several points throughout the work hint at this. However, there is also the consideration of intervention points 1,2,3 which seem odd as they points seen before any individual for whom fairness is considered is seen. That is, fairness about public individuals cannot be made there, independent of any other issues such as privacy budgeting. Further, Theorem 1 discusses a demographic parity pre-processor which achieves demographic parity on private data which I presume is irrelevant.
> - The statement "PATE relies on unlabeled public data, which lacks the ground truth labels Y" is a bit confusing unless one has already understood that fairness is with respect to public data. PATE also relies on private labeled data to create the teachers.
>
> 	- A fairness intervention and a privacy intervention have different goals w.r.t. the data distribution they target. In private learning, the goal is to protect the privacy of the sensitive training data only. In fair learning, the end goal is to ensure the algorithmic decisions (at inference-time) are fair. Within the context of FairPATE; from a private learning perspective, PATE (and FairPATE by extension) protect private teacher data (see Figure 1.a) and not the public unlabeld data. From a fair learning perspective, a FairPATE model is just like any other fairness-aware model is trained such that it achieves fairness on training data, **with the expectation that that behavior generalizes to inference-time.** These are standard assumptions which is why we have refrained from repeating them.
>
> Smallest things:
>
> - Double "violations" near "violations of demographic disparity violations".
> - The statement "DP that only protects privacy of a given sensitive feature" might be mischaracterizing DP. It is not focused on features or even data but rather the impact of *individuals* on visible results.
> 	- We thank the reviewer for their keen eye. We have fixed the typo and revised the sentence to
> 	  > ..DP that only protects privacy of individuals with respect to particular sensitive features...

---

> ### Author Response · Authors · 2023-11-18
> **Response to Questions**
>
> - Question A: Is reasonable to ignore downstream decisions from queries rejected due to fairness (i.e. contrary to my suggestion in the weaknesses above)?
>
> 	- We believe the alternative algorithm suggested is functionally similar to FairPATE. Please see our answer above.
>
> - Question B: C1 makes a point that adding privacy after fairness may break fairness. What about in expectation? Were one to view the demographic statistics defining fairness measures in expectations, wouldn't they remain fair?
>
> 	- The aforementioned fairness constraints are rate constraints which are in and of themselves expected values. If the constraints remain effective post DP noising; it means that we have retained at least group membership information. This is inconsistent with the standard neighboring relationship that our standard privacy definition adopts (i.e., one-sample difference between two datasets) therefore it is a weaker privacy notion. We conjecture that this weaker privacy notion would be closer to the DP w.r.t. sensitive attributes (which defines the group membership), but that it is possibly even weaker because prior theoretical work with DP w.r.t. sensitive attributes (namely, Mozannar et al. 2020) have already shown that a second post-processing step is necessary to ensure the fairness constraint is satisfied post DP-noising.
>
> - Question C: Theorem 1 makes a statement about a pre-processor inducing privacy parameter degradation but FairPATE (or PATE) appears to fit the definition of a pre-processor. If the point of the Theorem is to argue against pre-processors, isn't it also arguing against PATE/FairPATE? Unrelated, what is "ordering defined over the input space X" and why is it necessary?
>
> 	- Theorem 1 does not apply to FairPATE because it does not pre-process the training data as described in theorem 1. FairPATE is a pre-processor from the point of view of the student which only sees public queries. An algorithm that pre-processes the teacher data would indeed fit Theorem 1 because it sees private sensitive data (this is intervention point 1 in Figure 1.a).
>
> 	- The assumption on the ordering is a technicality that simplifies the the proof.  In practice, one can almost always assume such an ordering exists. An example of such ordering would be to order images based on their pixel values in some specified order of height, width and channel starting by checking the first pixel, then the second pixel, and so on.
>
>
> We once again thank the reviewer for their detailed feedback and kindly ask them to consider raising their score if we have addressed their concerns successfully. We are happy to answer further questions.

---

> ### Comment · Reviewer_Mex5 · 2023-12-05
>
> Thank you for the clarifications.
>
> - With regards to:
>
>   "From a fair learning perspective, a FairPATE model is just like any other fairness-aware model
>    is trained such that it achieves fairness on training data, with the expectation that that
>    behavior generalizes to inference-time. These are standard assumptions which is why we have
>    refrained from repeating them."
>
>   Some discussion there is warranted despite "standard assumptions". FairPATE (Algorithm 2 and
>   Algorithm 1) enforces fairness on the inference-time instances and I don't think this requires an
>   assumption of being distributed similarly to training-time data. The approach does not resemble
>   fairness-aware model training but rather an inference-time enforcement mechanism. The approach's
>   DP protection of inference-time instances is only coincidental and not the goal while fairness
>   w.r.t training-time instances is irrelevant as it is in any fairness training procedure
>   (decisions for training instances have already been made in the past as indicated by their
>   label). Is it accurate to say that "FairPATE addresses inference data fairness and training data
>   privacy"?
>
> - With regards to discussions about comparisons against baselines that do not have the option to
>   "reject due to fairness violation"; I don't see a suggestion of solution for this work and given
>   how close the results already were, my concerns regarding this fairness and results are not
>   addressed enough to warrant an upgrade in score large enough to the next threshold value.

---

### Official Review · Reviewer_XkhF · 2023-11-06

**Soundness:** 3 good
**Presentation:** 4 excellent
**Contribution:** 2 fair
**Rating:** 5
**Confidence:** 4

**Summary:**

The proposes a framework to integrate fairness into PATE. The proposed method is a simple adaptation of PATE which incorporates fairness constraints into the model's query rejection mechanism.

**Strengths:**

- The paper tackles a highly relevant issue in ML, addressing both theoretical and practical implications of fairness and privacy.
- The proposed framework is a simple adaptation of the existing PATE. Simplicity is a plus in my book.

**Weaknesses:**

- Fairness is "enforced" by adding parity constraint within the aggregator which acts as a rejection sampling step on the basis of fairness. The general idea was proposed in several other works in the past, including the ones cited by the authors and against which the authors compare. It is thus difficult to understand what is new in the proposed framework. An explicit mention would help.
- A discussion on when to use the proposed framework in contrast to other frameworks is absent (see also my questions below).
- The experimental analysis should be improved. Some figures are misleading, e.g., same colors used for different algorithms (see questions below).

**Questions:**

- I can't judge what is the impact of IPP (the rejection step added at inference time) on the overall results. Can you provide some ablation study showing how the framework performs on various datasets with different majority/minority distributions with and without IPP?
- How does the framework work in case of some distribution shift? This is especially important in the context of my question above.
- For the other datasets reported in the appendix the trends shown reverts, e.g., DP-Fermi produces better tradeoffs than FairPATE. Can you discuss why? What feature of the dataset makes this possible?
- Fig. 2 and 3 use orange colors for two different algorithms (Tran et al (Fig 2) and Jagielski et al. (Fig 3)). The authors should report all algorithms in all figures or justify their absence.
- Why Tran et and Jagielski et al. are not reported for the UTK-dataset experiment?
- Paper [Learning with Impartiality to Walk on the Pareto Frontier of Fairness, Privacy, and Utility](https://arxiv.org/pdf/2302.09183.pdf) discusses a similar topic (although the contributions from this work are different) and it could be added to your Related work section.

Minor comments:

A lot of the cited papers have appeared in conferences. But the authors cite their arxiv version. I suggest to update the references accordingly.

---

> ### Author Response · Authors · 2023-11-18
> **Response to Weaknesses**
>
> We thank the reviewer for their feedback. Below we address each point raised in "Weaknesses" section.
>
> - Fairness is "enforced" by adding parity constraint within the aggregator which acts as a rejection sampling step on the basis of fairness. The general idea was proposed in several other works in the past, including the ones cited by the authors and against which the authors compare. It is thus difficult to understand what is new in the proposed framework. An explicit mention would help.
> 	- The reviewer is correct that our work enforces fairness on the level of sample selection. However, this is done within a differentially private learning setup. The only prior work that we are aware of that has a sampling step for fairness within a private-learning setup is Kulynych et. al 2021 but importantly that work does not consider the privacy cost of the fairness importance sampling it performs while our work does.
> 	- We have added an extended related works section in Section I in the appendix with a more detailed comparison to Kulynych et. al 2021.
> - A discussion on when to use the proposed framework in contrast to other frameworks is absent (see also my questions below).
> 	- We have addressed the reviewer's concern. Please see our General Response.
> - The experimental analysis should be improved. Some figures are misleading, e.g., same colors used for different algorithms (see questions below).
> 	- We have re-generated the figures in question (3 and 4) and updated the manuscript accordingly.

---

> ### Author Response · Authors · 2023-11-18
> **Response to Questions**
>
> We kindly note that in the following, where appropriate, we have grouped the reviewer's questions together:
>
> - I can't judge what is the impact of IPP (the rejection step added at inference time) on the overall results. Can you provide some ablation study showing how the framework performs on various datasets with different majority/minority distributions with and without IPP?
> - How does the framework work in case of some distribution shift? This is especially important in the context of my question above.
> 	- We have run additional experiments on FairPATE+IPP and have added the results Section E.2 in the appendix of the updated manuscript. We observe that under class-imbalance for every percentage of coverage degradation, the fairness-privacy curve improves consistently. For the balanced case, we observe that the disparity levels are much higher. This is because random (non-stratified) sampling is not conducive to demographic parity.  IPP's overall behavior is similar as before but the curves are much closer to each other and the rejection rates are higher due to high disparity levels of the initial models caused by random sampling.
>
> - For the other datasets reported in the appendix the trends shown reverts, e.g., DP-Fermi produces better tradeoffs than FairPATE. Can you discuss why? What feature of the dataset makes this possible?
> 	- We thank the reviewer for the insightful question. Adult is a label-imbalanced dataset and we believe this is the reason its results stand out. Let us elaborate on this point: FairPATE achieves better fairness by pre-processing (a sample-level mitigation). As a result, if the dataset is imbalanced; especially if it is conditionally imbalanced, that is  $\mathbb{P}[Y = 1 \mid Z = 0] - \mathbb{P}[Y = 1 \mid Z = 1] > \gamma,$ then fairness of a model trained on such data would also be bounded from below: $\mathbb{P}[\hat{Y} = 1 \mid Z = 0] - \mathbb{P}[\hat{Y} = 1 \mid Z = 1] > \alpha > 0.$  This follows from Shamsabadi et al. 2023 Theorem 1, if we assume that the ground truth labels $Y$ are coming from a data oracle and the predicted labels $\hat{Y}$ are from the "surrogate model". Since the oracle model is not fair; then the actual model trained on those will similarly not be fair (in the demographic parity sense).
>
> 	- DP-FERMI on the other hand has a model-level mitigation with a fairness regularization term scaled with $\lambda \in \mathbb{R}^+$ . For a large enough $\lambda$ , DP-FERMI can close the fairness gap on a particular dataset. However, this comes at a cost to generalization. Prior works in algorithmic fairness have noted this trade-off under the notion of "stability" and "generalization of fairness" (see for instance, Huang and Vishnoi 2020). We note that in Section D in the appendix, we also present a weight-space mechanism  (namely, model fair-tuning) and show it to be similarly effective in reducing the residual fairness gap of a FairPATE student model.
>
>
> - Fig. 2 and 3 use orange colors for two different algorithms (Tran et al (Fig 2) and Jagielski et al. (Fig 3)). The authors should report all algorithms in all figures or justify their absence.
> 	- We have regenerated and replaced Figure 3 to ensure distinct colors. We apologize for the confusion.
> 	- As to the reason why Figure 3 does not include Tran 21 et al, we have already made an explicit note in the empirical section that we report the same  baseline results as reported by Lowy et al. 2023 (in 2nd line of SOTA Baseline Comparisons) including that of Tran et al 2021.
>
> - Why Tran et and Jagielski et al. are not reported for the UTK-dataset experiment?
>
> 	- Both methods are evaluated on tabular data and are not suitable for deep learning models. In particular, Lowy et al. 2023 report that they have attempted this to no avail:
> 	  > We observed that the baselines were very unstable while training and mostly gave degenerate results(predicting a single output irrespective of the input). By contrast, our method was able to obtain stable and meaningful tradeoff curves.
>
> 	  Given that DP-FERMI reports better fairness-accuracy trade-offs at every privacy level compared to these baselines; we found it sufficient to test FairPATE against the new state-of-the-art (DP-FERMI); and report the other baselines for which we had reported data.
>
> Minor comments:
>
> - A lot of the cited papers have appeared in conferences. But the authors cite their arxiv version. I suggest to update the references accordingly.
> 	 - We thank the reviewer for their careful reading of the paper. We have replaced the arxiv papers with conference counterparts.
>
> We once again thank the reviewer for their detailed feedback and kindly ask them to consider raising their score if we have addressed their concerns successfully. We are happy to answer further questions.

---

> > ### Comment · Reviewer_XkhF · 2023-11-22
> > **Response**
> >
> > Thank you for your replies. They are appreciated.
> > I suggest to make it much more explicit the point discussed here regarding distribution shift and trends regarding different datasets, as the current results showcased (if one would read the main section only) may be misleading or at least report only a portion of the story.
> >
> > I also strongly suggest the authors to include the evaluation of Mozannar et al. al and Tran et al. in all of the experiments (including those based on CNNs)!
> > In particular, notice that the claim by Lowey et al. is erroneous and both algorithms are reported on the UTK datasets in their original papers:
> > - [Tran and Fioretto, On the Fairness Impacts of Private Ensembles Models](https://www.ijcai.org/proceedings/2023/0057.pdf) See figure 7 (right)
> > - [Tran et al, SF-PATE: Scalable, Fair, and Private Aggregation of Teacher Ensembles](https://www.ijcai.org/proceedings/2023/0056.pdf) See figure 2(c)

---

> > > ### Author Response · Authors · 2023-11-23
> > > **Thank you for your time and suggestions!**
> > >
> > > We thank the reviewer for engaging with us. We appreciate the time and effort you have dedicated to reviewing our work.
> > >
> > > - Thank you for your replies. They are appreciated. I suggest to make it much more explicit the point discussed here regarding distribution shift and trends regarding different datasets, as the current results showcased (if one would read the main section only) may be misleading or at least report only a portion of the story.
> > >
> > > 	- We note that **we had already mentioned the particular strengths of our methods very prominently in the abstract and the introduction** as well as the experimental section. **Our method does better under Equality of Odds.** Under Demographic Parity it is a toss-up between the two methods as we had originally mentioned just below figures 2 and 3 in paragraph SOTA Baseline Comparisons.
> > >
> > > 	- We nevertheless understand that the placement of one of the figures in the appendix had made comparisons difficult, so **we have brought up the demographic parity results on Adult in Figure 2**. We have also regenerated the figure with better-tuned parameters which gives tighter confidence intervals.
> > >
> > > - I also strongly suggest the authors to include the evaluation of Mozannar et al. al and Tran et al. in all of the experiments (including those based on CNNs)! In particular, notice that the claim by Lowey et al. is erroneous and both algorithms are reported on the UTK datasets in their original papers:
> > >   [Tran and Fioretto, On the Fairness Impacts of Private Ensembles Models](https://www.ijcai.org/proceedings/2023/0057.pdf) See figure 7 (right)
> > >   [Tran et al, SF-PATE: Scalable, Fair, and Private Aggregation of Teacher Ensembles](https://www.ijcai.org/proceedings/2023/0056.pdf) See figure 2(c)
> > > 	- The official repository of Mozannar et al. 2020 [solves a constrained linear program](https://github.com/husseinmozannar/fairlearn_private_data/blob/master/adult_experiment.ipynb) which is inappropriate for vision datasets. If the reviewer is aware of any other implementations, we would be grateful if they shared it with us. Also, as far as we know there is no published code base for any of the two Tran papers mentioned. If the reviewer is aware of such code bases, we would be grateful if they shared it as a part of the meta-review.
> > >
> > > 	- We note that "On the Fairness Impacts of Private Ensembles Models" adopt **excessive risk** as their fairness metric "which is defined as the difference between the private and non-private risk functions." This is a very different type of fairness metric that is incompatible with the group fairness metrics that we, Lowy et al, or even the other Tran paper proposed here (SF-PATE) adopts. Furthermore, as we elaborated in the general response (and the table), SF-PATE adopts DP-w.r.t. Sensitive Attributes instead of the general $(\epsilon, \delta)$ -DP. Therefore, given the incompatible definitions of fairness and privacy, we do not believe these are good baselines for our methods.

---

> > > > ### Comment · Reviewer_XkhF · 2023-12-03
> > > > **Response**
> > > >
> > > > Dear authors, thanks a lot for your response. It clarified some of my doubts.

---

### Author Response · Authors · 2023-11-18
**General Author Response**

We thank the reviewers wholeheartedly for their detailed comments and constructive feedback. We are glad that all reviewer found the paper to be well-written and easy to follow, and that they appreciated the simplicity of our joint fairness-privacy rejection mechanism of FairPATE. One reviewer noted the potential of extending this mechanism to other [trustworthiness] contexts, while another appreciated our discussion of interventions point in PATE.

**New Manuscript and Color Codings.** We have applied reviewer's comments within the manuscript and have highlighted newly added parts in "blue" and fixed typos in "red". In order to fit the new manuscript within 9 pages we have also shortened certain sections of the paper which are marked in "teal."

We will shortly address a common reviewer's question regarding comparison to prior work and provide a table of comparisons for methods to which our work has conceptual similarities but which are ultimately unsuitable for a fair empirical comparison. This is due to a difference in privacy notion, or lack of privacy accounting for privacy leakage of the fairness intervention.


**Contribution & Guidance on Choice of Framework**

The contribution of our algorithm is not limited to performance  improvements over our main baseline Lowy et al. 2023 (although we also demonstrate those). As we discuss in Sections 6, prior work has largely adopted a limited privacy notion for joint private-and-fair learning; namely that of **DP w.r.t. sensitive attributes Jagielski et al. 2019**. While this notion has its particular applications (e.g. for example to train ethnically-fair models that also do not reveal the ethnicity of the individual), it does not preclude catastrophic failures of privacy that more general notions of differential privacy protects against—in the previous example, for an ethnic individual's medical features to be leaked as a result of release of the aforementioned model.

The main contribution of our work is to enable practical and $(\varepsilon,\delta)$ -DP training (i.e., with respect to all features and not just the sensitive ones) in a PATE-type framework; while Lowy et al. 2023 is the first work to enable this using a DP-SGD-type framework. We note that there are benefits and draw-backs to using either PATE and DP-SGD frameworks which by extension makes both FairPATE and DP-FERMI viable in their own regard. We have highlighted some of these trade-offs in Section 7.

Furthermore, we have added an extended related works in Section I in the appendix, and have added the following table and paragraph on guidance for choice of framework to Section I.2.

| **Method** | $(\epsilon, \delta)$ **-DP** | **DP w.r.t. Sensitive Attributes** | **Privacy Accounting for Fairness** | **Public Data** | **Fairness Intervention Space** |
|---|---|---|---|---|---|
| **FairPATE** | $\checkmark$ | - | $\checkmark$ | Unlabeled | Samples |
| **DP-FERMI** | $\checkmark$ | - | $\checkmark$ | - | Weights |
| **SF-PATE** | - | $\checkmark$ | $\checkmark$ | Unlabeled | Weights |
| **Jagielski et al. 2019** | - | $\checkmark$ | $\checkmark$ | - | Weights |
| **Mozannar et al. 2020** | - | $\checkmark$ | $\checkmark$ | - | Weights |
| **DP-IS SGD** | $\checkmark$ | - | $\checkmark$ | - | Samples |
| **Zhang et al. 2021** | $\checkmark$ | - | ✗ | - | Opt. Hyperparams |

**Choice of Framework.** The appropriate choice of privacy notion is application-dependent. However, a $(\epsilon, \delta)$ -DP is a much stronger guarantee and it is by far the most prevalent notion of privacy. If interpretability is of concern, sample-space interventions are more appropriate since reasoning about the membership (or probability of membership) of a certain sample is easier than reasoning about high-dimensional gradients in the weight space. Finally, all PATE-based frameworks assume access to an unlabeled public dataset due to its semi-supervised nature while DP-SGD-type methods do not require such dataset. This need can be addressed by using a subset of the training data, throwing out its labels, and employing it in place of the unlabeled dataset in PATE-based models. This is how the original PATE papers were evaluated, and we have evaluated FairPATE similarly.

In practice, however, the unlabeled dataset can come from a public dataset which is different than the sensitive training set. From a privacy accounting perspective only, the use of such public data is similar to the use of public data to pre-train DP-SGD models: in both cases, privacy cost, by definition, is not being accumulated for the public data. The difference is that pre-training requires labels, while the PATE's public dataset need not be labeled.

We are happy to further engage with reviewers as part of the discussion phase and hope that the reviewers consider raising their scores.

__Edits__ This comment has been edited for clarity after interactions with reviewer p5Rv. We thank the reviewer for their comment.

---

> ### Comment · Reviewer_p5Rv · 2023-11-18
>
> Dear Authors,
>
> Thanks for the response. Just to clarify when you say
>
> '''Finally, all PATE-based frameworks assume access to an unlabeled public dataset on which privacy accounting does not take place. This is akin to pre-training DP-SGD models on public data before any privacy accounting occurs—although this requires labels while PATE-based models do not.'''
>
> __Do you mean PATE uses the same kind of data, but without labels, as DP-SGD normally uses for pre-training ?__
>
> From my understanding, the most popular examples of DP-SGD using pre-training is ImageNet pre-trained for private fine-tuning on CIFAR or JFT pre-training for fine-tuning on ImageNet and similar for NLP tasks.
>
> Is your public data from a very different distribution compared to the test set ? That's not the impression I got from reading the draft.

---

> > ### Author Response · Authors · 2023-11-18
> > **Thank you for the question**
> >
> > In evaluating PATE-type models against DP-SGD-type models, it is customary to use a split of the training data, throw out its labels and use it to query the aforementioned teacher ensemble. This ensures that PATE does not observe more information than their DP-SGD counterparts; and it is how the two original PATE papers were evaluated.
> >
> > So the answer to the highlighted question, in the context of our evaluations, is no. We use a subset of the training data.
> >
> > Our prior comment was meant as an analogy for the privacy accounting only: pre-training data is not accounted for in privacy budgeting of DP-SGD; just as the unlabeled data is not accounted for in privacy budgeting of PATE. In other regards (distribution, etc.) the pre-training data, and the unlabeled dataset are not similar.

---

> > > ### Comment · Reviewer_p5Rv · 2023-11-22
> > >
> > > Thanks for the clarification. I think the way the above (original comment) statement is made is misleading and I would suggest that the authors update it to reflect what is mentioned in the clarification.

---

> > > > ### Author Response · Authors · 2023-11-22
> > > >
> > > > We have applied your suggestion and edited the response for clarity on OpenReview and also in the paper. Note that we will update the latter on OpenReview after we have also applied other reviewers' additional comments.
> > > >
> > > > Thank you for interacting with us!

---

### Author Response · Authors · 2023-11-21
**Thank you for your feedback. Please consider our rebuttal responses.**

Dear reviewers,

Thank you for your valuable feedback. We believe it has improved our submission.  Given that the author interaction window is coming to an end, we would like to kindly ask you to consider the rebuttal responses and let us know if they address your concerns and, if they do, to consider raising your scores.

It would be our pleasure to answer any further questions.

---

### Meta-Review · Area_Chair_q9mW · 2023-12-10

**Metareview:**

The submission introduces FairPATE, an enhancement of the PATE framework that integrates fairness constraints. FairPate ensures more accurate differential privacy (DP) accounting by considering the potential for query rejections based on fairness. Numerical results demonstrate favorable privacy-fairness-accuracy trade-offs, outperforming prior methods in terms of fairness and accuracy for a given privacy budget.

Overall, the paper is clearly written, and the results are promising. However, the reviewers highlighted several limitations that must be addressed before acceptance.

Reviewer Mex5 highlighted the need to consider fairness when rejecting queries in privacy models, noting that ignoring fairness can be both consequential in practice and, in experiments, impact the Pareto frontier's observed gains. They also suggested the importance of including human judgments in the fairness calculus, even when a model refuses to answer a query due to fairness concerns. Such guidelines could improve the positive impact of FairPate.

Reviewer XkhF noted a disconnection with prior work. I read the authors' comments in this regard, and note their effort in addressing this issue. However, the reviewer's concerns remained post-rebuttal.

Finally, reviewer p5Rv noted the small improvements of FairPATE over baselines and raised concerns about the diversity and clarity of experiments, with several benchmarks relegated to the appendix without thorough discussion. The reviewer also challenged comparisons with Loewy et al.'s algorithm, arguing that the former benefits from public unlabelled data while the latter adheres to stricter privacy constraints. They also raised issues with comparisons to related work.

Overall, this is a promising paper. However, given the many issues raised by the reviewers that persisted after the extensive discussion, I encourage the authors to revise their manuscript and re-submit.

**Justification For Why Not Higher Score:**

See the issues raised above, most related to concerns regarding prior work, presentation of results, and implications of the rejection mechanism in fairness in practice.

**Justification For Why Not Lower Score:**

N/A

---

### Decision · Program_Chairs · 2024-01-16

Reject